# Suppressor of fused controls perinatal expansion and quiescence of future dentate adult neural stem cells

Hirofumi Noguchi[1], Jesse Garcia Castillo[1], Kinichi Nakashima[2], Samuel J Pleasure[3]*

[1]Department of Neurology, University of California, San Francisco, San Francisco, United States; [2]Department of Stem Cell Biology and Medicine, Graduate School of Medical Sciences, Kyushu University, Fukuoka, Japan; [3]Programs in Neuroscience and Developmental Biology, Eli and Edythe Broad Center of Regeneration Medicine and Stem Cell Research, University of California, San Francisco, San Francisco, United States

**Abstract** Adult hippocampal neurogenesis requires the quiescent neural stem cell (NSC) pool to persist lifelong. However, establishment and maintenance of quiescent NSC pools during development is not understood. Here, we show that Suppressor of Fused (Sufu) controls establishment of the quiescent NSC pool during mouse dentate gyrus (DG) development by regulating Sonic Hedgehog (Shh) signaling activity. Deletion of *Sufu* in NSCs early in DG development decreases Shh signaling activity leading to reduced proliferation of NSCs, resulting in a small quiescent NSC pool in adult mice. We found that putative adult NSCs proliferate and increase their numbers in the first postnatal week and subsequently enter a quiescent state towards the end of the first postnatal week. In the absence of Sufu, postnatal expansion of NSCs is compromised, and NSCs prematurely become quiescent. Thus, Sufu is required for Shh signaling activity ensuring expansion and proper transition of NSC pools to quiescent states during DG development.

DOI: https://doi.org/10.7554/eLife.42918.001

*For correspondence:
sam.pleasure@ucsf.edu

**Competing interests:** The authors declare that no competing interests exist.

## Introduction

Newborn neurons are generated in two restricted regions of the adult rodent brain: the cortical subventricular zone (SVZ) and the dentate subgranular zone (SGZ) (*Altman and Das, 1965*; *Eriksson et al., 1998*; *Kuhn et al., 1996*; *Lois and Alvarez-Buylla, 1993*). Adult neurogenesis in the dentate gyrus (DG) has been implicated in hippocampal-dependent memory and learning (*Deng et al., 2010*). Newly generated neurons produced from neural stem cells (NSCs) residing in the SGZ are constantly added to the granule cell layer (GCL) and integrated into the existing hippocampal circuitry (*Imayoshi et al., 2008*). Persistence of adult hippocampal neurogenesis relies on the proper maintenance of NSCs even after development. However, little is known about the developmental programs governing the production and maintenance of long-lived NSCs.

Quiescence of NSCs during early development has been proposed to play a key mechanism for maintaining the NSC pool throughout life (*Furutachi et al., 2013*; *Kawaguchi et al., 2013*; *Mira et al., 2010*; *Song et al., 2012*). NSCs enter a quiescent state in a spatiotemporal manner during development, and this step is critical for ensuring the appropriate sized NSC pool for adult neurogenesis. At the beginning of forebrain development, NSCs are highly proliferative, but gradually lose proliferation competence with development and enter a quiescent state (*Furutachi et al., 2015*). The failure to transition to a quiescent state during development triggers continuous

proliferation of NSCs and leads to premature exhaustion of the NSC pool (*Furutachi et al., 2015*). Furthermore, the NSC pool must be properly established since NSCs can only undergo a limited number of rounds of cell division prior to terminal differentiation. Live imaging of NSCs in adult DG and thymidine-analog based cell tracing analysis demonstrated that quiescent NSCs undergo a series of asymmetric divisions to produce neurons, and subsequently are consumed by symmetric differentiation into astrocytes or neurons (*Calzolari et al., 2015*; *Encinas et al., 2011*; *Pilz et al., 2018*). Indeed, adult neurogenesis and the NSC pool have been shown to decline with aging (*Kuhn et al., 1996*; *Lugert et al., 2010*), suggesting that for neurogenesis to persist throughout life, the size of putative quiescent NSC pool must be established during development before the NSCs transition to a quiescent state.

Long-lived NSCs in the DG originate from the ventral hippocampus at E17.5 (*Li et al., 2013*). These cells migrate along the longitudinal axis of the hippocampus from the temporal to septal poles and eventually settle in the ventral and dorsal DG. The initial production and maintenance of long-lived NSCs is dependent on Shh signaling and Shh ligands, produced by local neurons in the embryonic amygdala and the postnatal DG (*Li et al., 2013*). Blocking Shh signaling by deleting *Smoothened (Smo)* from responsive cells in the DG or ablation of Shh ligands from local neurons impairs the emergence of long-lived NSCs and results in diminishing the NSC pool (*Han et al., 2008*; *Li et al., 2013*). These findings highlight the significance of Shh signaling in production of the NSC pool during development. What is not clear yet from these studies is how Shh signaling activity is spatiotemporally regulated to ensure the expansion of the NSC pool during DG development and the role of Shh signaling in the transition of NSCs to a quiescent state.

Shh signaling is critical at early stages of embryonic brain development. Thus, complete ablation of Shh signaling activity by *Smo* deletion or the constitutive activation of Shh signaling by expressing an active Smo mutant (SmoM2) severely compromise the initial steps of DG development (*Han et al., 2008*). The embryonic nature of this phenotype prevents the further analysis of specific roles of Shh signaling in postnatal DG development, particularly in the production and maintenance of postnatal NSCs. To circumvent this, we are utilizing a Cre-loxP based system that allows spatio-temporal analysis of Shh signaling activity by genetic manipulation of the Shh signaling inhibitor, Suppressor of Fused (Sufu), a Gli-binding protein with an indispensable role in embryonic development. Conditional deletion of Sufu in a spatiotemporal manner allowed us to examine the role of Shh signaling in various aspects of NSC behavior during DG development. Our earlier studies showed that Sufu is important for the specification of NSC fate decision during cortical development via regulating Shh signaling activity (*Yabut et al., 2015*). In this report, we set out to determine the contribution of Sufu in regulating Shh signaling during DG development and how Sufu and Shh signaling are involved in the mechanisms governing the expansion of long-lived NSCs and their transition to the quiescent state during DG development. Intriguingly, we find that deletion of *Sufu* decreases Shh signaling in NSCs during DG development – this is in distinction to the neocortex where loss of *Sufu* increases Shh signaling. Long-lived NSCs expand in the early part of first postnatal week, but proliferation of these NSCs is impaired in the absence of Sufu, resulting in a decreased NSC pool in the adult DG. We also found that long-lived NSCs gradually become quiescent towards the end of the first postnatal week. However, *Sufu* deletion precociously triggers this transition to the quiescent state. Taken together, these results indicate that loss of Sufu during DG development decreases Shh signaling activity and impairs expansion of long-lived NSCs and the timely transition to a quiescent state during DG development.

## Results

### Deletion of *Sufu* in NSCs reduces Shh signaling during DG development

Shh ligands originate from amygdala neurons and the adjacent ventral dentate neuroepithelium to activate Shh signaling in ventral hippocampal NSCs (*Li et al., 2013*). These Shh-responding NSCs subsequently migrate to the dorsal DG and gradually accumulate between the hilus and GCL to form the SGZ postnatally (*Li et al., 2013*). Sufu is expressed in NSCs of the developing forebrain including presumptive DG cells (*Yabut et al., 2015*). We previously reported that deletion of *Sufu* in NSCs at early gestational stages (E10.5) severely disrupted the overall cytoarchitecture of the forebrain as a consequence of ectopic activation of Shh signaling (*Yabut et al., 2015*). To determine the

effects of Sufu specifically in DG development, we used a hGFAP-Cre line to delete *Sufu* in NSCs at E13.5, before the initiation of DG development (E14.5) – we call these mice hGFAP-Sufu-KO. We first asked if deletion of *Sufu* increases Shh signaling activity. Expression of *Gli1*, a downstream target gene of Shh signaling, is strictly dependent on Shh signaling stimulation. Thus, we used Gli1-LacZ reporter mice (*Bai et al., 2002*), in which lacZ is expressed under the Gli1 promoter, to identify and characterize Shh-responding cells in the developing DG of hGFAP-Sufu-KO mice. Accordingly, we found abundant Gli1-lacZ + cells in the ventricular zone of the ventral hippocampus in *Sufu*^*fl/fl*^; *Gli1*^*lacZ/+*^ mice (*Figure 1A*). Gli1-lacZ + cells were also present in the dorsal DG of *Sufu*^*fl/fl*^;*Gli1*^*lacZ/+*^- mice at P0 and were enriched in the SGZ at P7 (*Figure 1B–C* and *Figure 1—figure supplement 1*). Surprisingly, we found a remarkable reduction of Gli1-LacZ + cells in ventral hippocampus of hGFAP-Sufu-KO;Gli1^lacZ/+^ mice at E15.5-P0, and small numbers detected throughout the anterior to posterior DG at P7. These data demonstrate that deletion of *Sufu* in NSCs decreases Shh signaling activity during DG development. This is distinct from the embryonic neocortex where deletion of Sufu increases Shh signaling activity (*Yabut et al., 2015*).

## Deletion of Sufu decreases proliferation of NSCs in developing DG

Shh signaling plays a pivotal role in establishing and maintaining the NSC pool to adulthood (*Choe et al., 2015*; *Han et al., 2008*; *Li et al., 2013*). Ablation of Shh signaling in NSCs by deleting *Smo* leads to a drastic reduction in NSC proliferation and results in the failure of SGZ establishment (*Han et al., 2008*; *Li et al., 2013*). Since hGFAP-Sufu-KO;Gli1^lacZ/+^ mice showed a reduction in Gli1 expression at the onset of DG development, we investigated if deletion of *Sufu* influences the proliferation capacity of NSCs particularly in the SGZ where Sox2 +cells form the NSC pool. During early postnatal DG development, NSCs migrate to the border between the hilus and GCL to form SGZ. The SGZ can be distinguished by accumulation of Sox2 +cells around P7 and is fully established by second postnatal week. Thus, we examined the number of Sox2 +cells in SGZ at P7 and P14. Although we found no difference in the number of Sox2 +cells between Sufu^fl/fl^ mice and hGFAP-Sufu-KO mice in the SGZ (*Figure 2A and B*), there was a significant reduction in Ki67 +proliferating cells in the P7 hGFAP-Sufu-KO mice (*Figure 2C*). In addition, there was a significant reduction in ratio of Ki67 +cells to Sox2 +NSCs in the SGZ of hGFAP-Sufu-KO mice (*Figure 2D*) indicating that very few Sox2 +cells retained their proliferative capacity. We next examined the number of Sox2 +cells and Ki67 proliferating cells in the SGZ of DGs at P14. In addition to the reduction in Ki67 +proliferating cells, the number of Sox2 +cells was significantly reduced in SGZ of hGFAP-Sufu-KO mice at P14 (*Figure 2E–G*). These data suggest that deletion of *Sufu* decreases NSC proliferation and number. We found that these phenotypes of Sufu-KO mice contrasts starkly with those observed in the hGFAP-Cre;SmoM2 mice, in which Shh signaling is constitutively active in NSCs. The DG of hGFAP-Cre;SmoM2 mice display abnormal morphology (*Figure 2H*) as previously reported (*Han et al., 2008*) and exhibit higher numbers of dying cells as marked by cleaved caspase 3 + cells (*Figure 2—figure supplement 1A–C*). Despite this, there was a significant increase in proliferating Ki67 +cells in the SGZ of hGFAP-Cre;SmoM2 compared to WT mice at P7 (*Figure 2I and J*). Accordingly, the proliferating population of Sox2 +cells was significantly higher than WT mice at P7 (*Figure 2K*). Taken together, these findings establish the crucial roles of Shh signaling in regulating NSC proliferation in the developing DG. Importantly, that deletion of *Sufu* resulted in downregulation, instead of ectopic activation, of Shh signaling in DG NSCs, indicates the distinct effects of Sufu in specific NSC subtypes and may involve a novel mechanism by which Sufu controls Shh signaling activity in DG NSCs.

## In the absence of Sufu, Gli1 function becomes responsible for proper proliferation of NSCs during DG development

Gli1, Gli2 and Gli3 are the main transcription factors that transduce Shh signaling to downstream targets. Gli3 mainly functions as a gene repressor (*Hu et al., 2006*; *Litingtung et al., 2002*; *Persson et al., 2002*; *Wang et al., 2014*), whereas Gli1 and Gli2 are responsible for activating target gene expression (*Bai and Joyner, 2001*; *Park et al., 2000*). Gli2 and Gli3 play major roles in regulating gene expression during the development, whereas Gli1 is largely dispensable (*Bai et al., 2002*; *Park et al., 2000*). Indeed, in the developing DG, we did not observe any differences in the number of Sox2 +cells and proliferating Sox2 +NSC in the SGZ between *Sufu*^*fl/fl*^, *Sufu*^*fl/fl*^;*Gli1*^*lacZ/+*^ and *Sufu*^*fl/*^

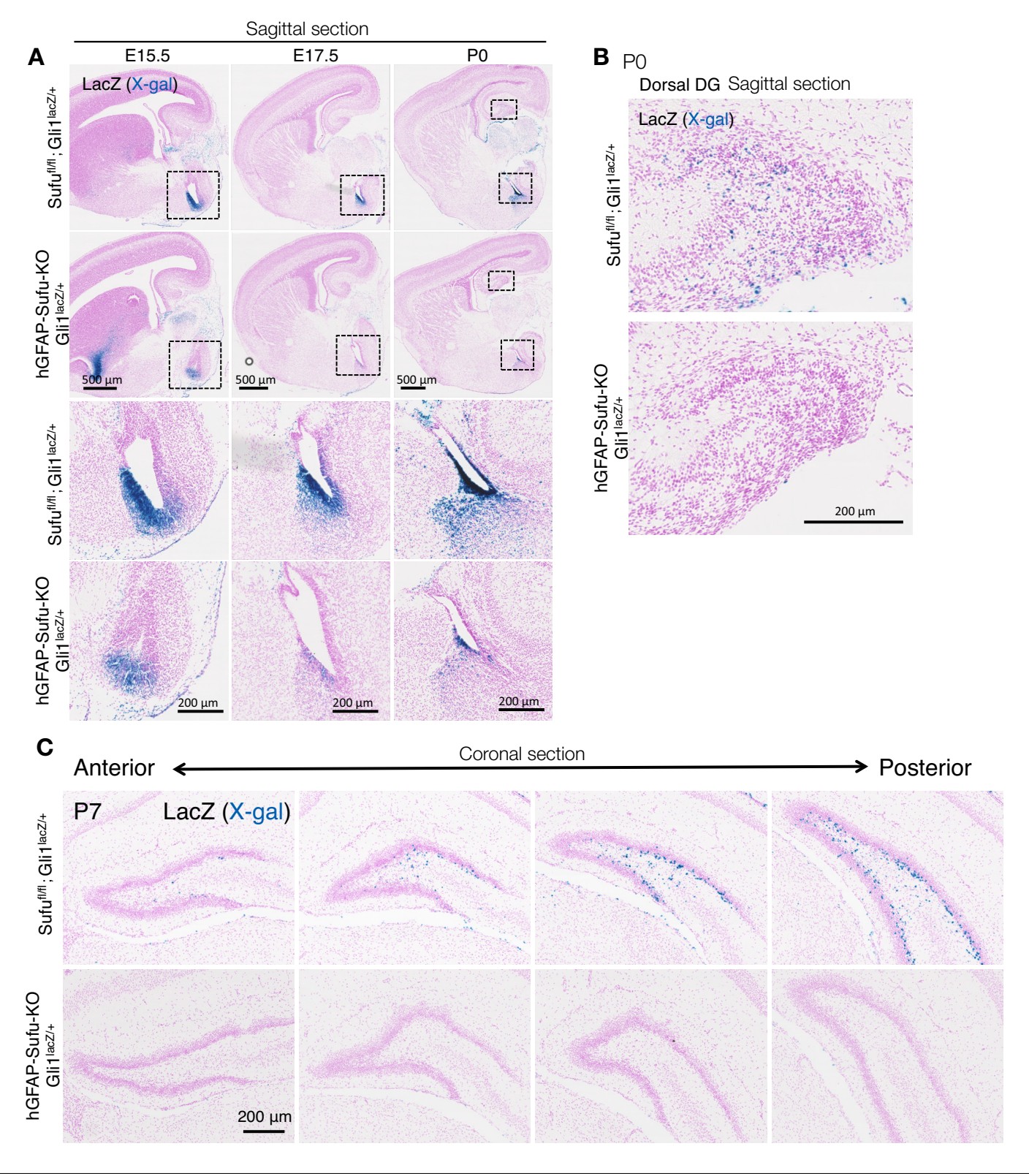

**Figure 1.** Deletion of *Sufu* decreases Hh-responding cells during DG development. (A) Representative Gli1-LacZ staining images of sagittal brain sections in *Sufu^{fl/fl};Gli1^{lacZ/+}* and hGFAP-Sufu-KO;Gli1^{lacZ/+} mice at E15.5-P0. Magnified images of the black dashed-line boxes are shown to below of each image. (B) Representative Gli1-LacZ staining images of dorsal DG in sagittal sections of *Sufu^{fl/fl};Gli1^{lacZ/+}* and hGFAP-Sufu-KO;Gli1^{lacZ/+} mice. (C)

*Figure 1 continued on next page*

*Figure 1 continued*

From anterior to posterior, four levels of coronal sections for Gli1-nLacZ staining at E17.5 are shown. Note that lacZ +cells are diminished in the DG of hGFAP-Sufu-KO;Gli1$^{lacZ/+}$ mice from the beginning of DG development.

DOI: https://doi.org/10.7554/eLife.42918.002

The following figure supplement is available for figure 1:

**Figure supplement 1.** Deletion of *Sufu* decreases Hh-responding cells during DG development.

DOI: https://doi.org/10.7554/eLife.42918.003

$^{fl}$;Gli1$^{lacZ/lacZ}$ mice (***Figure 3—figure supplement 1A–D***). Therefore, we decided to use those genotypes as control. To our surprise, however, we found that deletion of one (hGFAP-Sufu-KO;Gli1$^{lacZ/+}$) or both (hGFAP-Sufu-KO;Gli1$^{lacZ/lacZ}$) *Gli1* alleles in hGFAP-Sufu-KO mice led to a more profound phenotype. Overall, Ki67 +cells and the proliferating population of Sox2 +cells in the SGZ were significantly reduced in hGFAP-Sufu-KO;Gli1$^{lacZ/+}$ and hGFAP-Sufu-KO;Gli1$^{lacZ/lacZ}$ mice compared with control at P7 (***Figure 3A–C***). In addition, we observed changes in NSC populations; hGFAP-Sufu-KO;Gli1$^{lacZ/+}$ and hGFAP-Sufu-KO;Gli1$^{lacZ/lacZ}$ mice displayed a remarkable decline in Sox2 +cells in contrast with hGFAP-Sufu-KO mice at P7 (***Figure 3D***). These differences remained at P14 (***Figure 3—figure supplement 2A–C***). Given that the number of Sox2 +cells were comparable in hGFAP-Sufu-KO mice at P7 whereas significantly decreased in hGFAP-Sufu-KO;Gli1$^{lacZ/+}$ mice, these findings indicate that deletion of *Gli1* allele further impairs the proliferation of NSCs in GFAP-Sufu-KO mice.

During the first postnatal week, NSCs give rise to neuronal precursor cells, which differentiate into granule neurons to form the GCL. Since neuronal precursor cells were produced from NSCs when they divide, we next examined if the reduction in proliferating Sox2 +cells correlated with a decrease in neuronal production. We found that Tbr2 +neuronal precursor cells, which was significantly reduced in hGFAP-Sufu-KO mice, was further reduced in hGFAP-Sufu-KO;Gli1$^{lacZ/+}$ mice at P7 and P14 (***Figure 3E,F*** and ***Figure 3—figure supplement 2D and E***). Since neuronal precursors are also proliferative, we wondered if the reduced number of Tbr2 +cells in hGFAP-Sufu-KO and hGFAP-Sufu-KO;Gli1$^{lacZ/+}$ mice resulted from decreased proliferation of neuronal precursors. However, we found no difference in Ki67 +population of neuronal precursors between control, hGFAP-Sufu-KO and hGFAP-Sufu-KO;Gli1$^{lacZ/+}$ mice (***Figure 3—figure supplement 2F and G***). This suggests that deletion of *Sufu* does not impair the proliferation of neuronal precursors, and that impaired proliferation of NSCs leads to reduction of Tbr2 +cells in hGFAP-Sufu-KO and hGFAP-Sufu-KO;Gli1$^{lacZ/+}$ mice. Together, these data suggest that deletion of *Sufu* impairs neuronal production in postnatal DG development, and this effect worsens when *Gli1* expression is reduced.

The profound phenotypes of hGFAP-Sufu-KO;Gli1$^{lacZ/+}$ mice compared with hGFAP-Sufu-KO suggest that deletion of *Sufu* increases the requirement for Gli1 function to regulate Shh signaling activity in DG development. To confirm this, we investigated the expression of Gli transcription factors transducing Shh signaling and their target genes. We extracted RNA from P3 DGs and examined the expression of *Gli1*, *Gli2* and *Gli3* by qPCR. As expected, *Gli1* expression was reduced in the DG of hGFAP-Sufu-KO;Gli1$^{lacZ/+}$ mice (***Figure 3G***) and correlated with the reduction of Gli1-lacZ + cells (***Figure 1***). Interestingly, these mice also showed remarkable reduction in *Gli2* and Gli3 expression (***Figure 3G***). The reduction of both Gli activators (Gli1 and Gli2) in hGFAP-Sufu-KO;Gli1$^{lacZ/+}$ mice raise the possibility that reducing *Gli1* in mice lacking *Sufu* further decreases Shh signaling activity in NSCs. This assumption is in line with previous reports of severe developmental defects when Gli2 levels are reduced in Gli1 homozygous null mutant mice (***Bai et al., 2002***; ***Park et al., 2000***). To confirm reduced Shh signaling activity, we also checked the expression of Shh signaling target genes, *Ptch1, N-myc, Cyclin D1* and *Cxcr4* (***Inaguma et al., 2015***; ***Stecca and Ruiz i Altaba, 2005***; ***Yin et al., 2019***); ***Zheng et al., 2013***), and found significant reductions in the expression of Shh signaling target genes in hGFAP-Sufu-KO;Gli1$^{lacZ/+}$ mice (***Figure 3—figure supplement 2H***). Taken together, these data suggest that *Sufu* deletion increases the dependency on Gli1 function during DG development, and that proliferation of NSCs is severely impaired when *Gli1* expression is reduced in absence of Sufu.

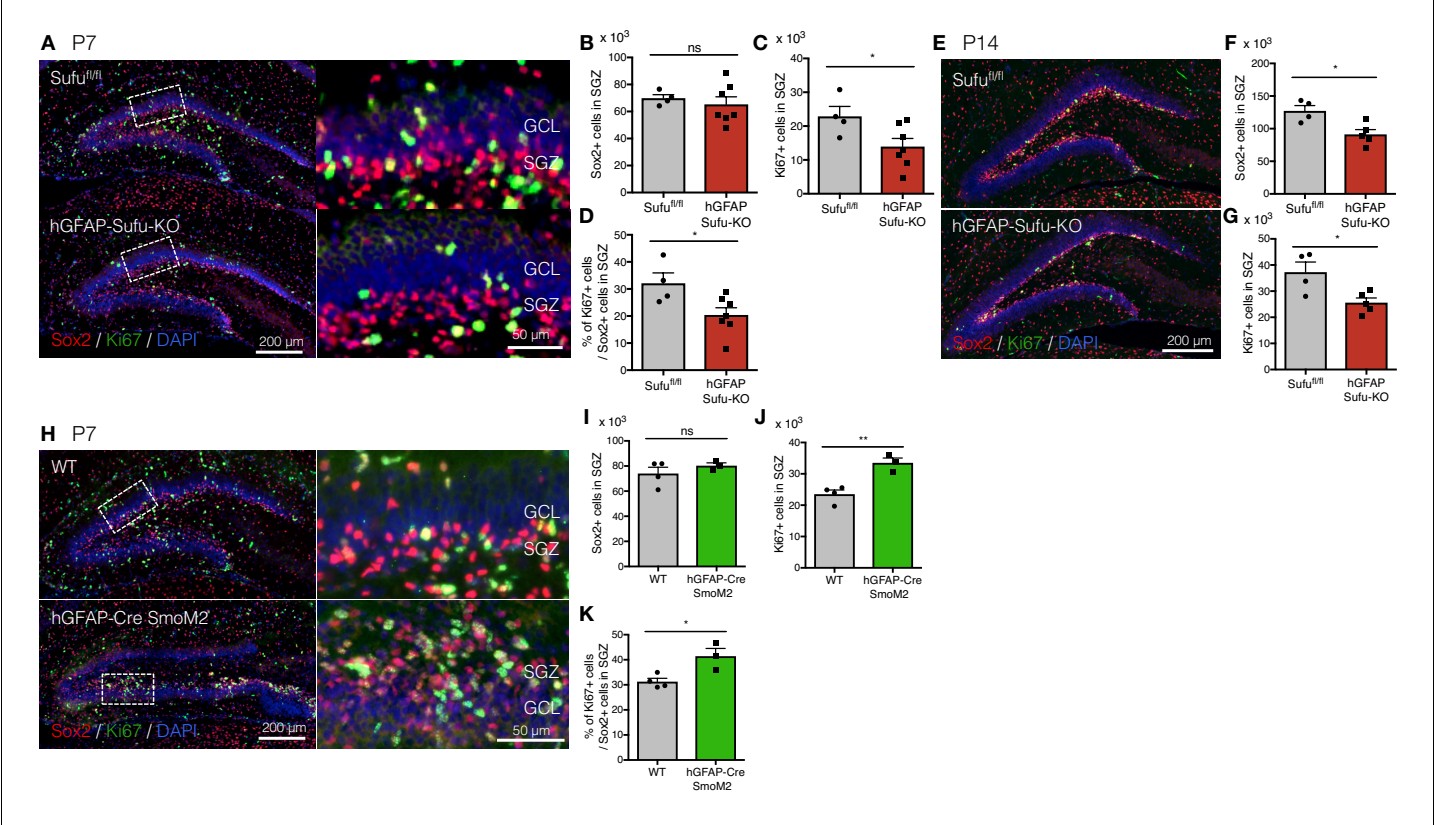

**Figure 2.** Deletion of *Sufu* decreases proliferation of DG-NSCs. (**A**) Representative immunofluorescence images of Sox2 (red) and Ki67 (green) in the DG of *Sufu^fl/fl* and hGFAP-Sufu-KO mice at P7. DNA is stained with DAPI (blue). Magnified images of the white dashed-line boxes are shown to the right of each image. (**B,C**) Quantification of Sox2+ (**B**) and Ki67+ (**C**) cells in the SGZ [*Sufu^fl/fl*, n = 5; hGFAP-Sufu-KO n = 7]. (**D**) The bar graph indicates the ratio of Ki67 +cells to Sox2 +NSCs in the SGZ [*Sufu^fl/fl*, n = 5; hGFAP-Sufu-KO n = 7]. (**E**) Representative immunofluorescence images of Sox2 (red) and Ki67 (green) in the DG of *Sufu^fl/fl* and hGFAP-Sufu-KO mice at P14. (**F, G**) Quantification of Sox2+ (**F**) and Ki67+ (**G**) cells in the SGZ [*Sufu^fl/fl*, n = 4; hGFAP-Sufu-KO n = 5]. (**H**) Representative immunofluorescence images of Sox2 (red), Ki67 (green) and DAPI (blue) in the DG of WT and hGFAP-Cre; SmoM2 mice at P7. Magnified images of the white dashed-line boxes are shown to the right of each image. (**I,J**) Quantification of Sox2+ (**I**) and Ki67+ (**J**) cells in the SGZ [WT, n = 4; hGFAP-Cre;SmoM2, n = 3]. (**K**) The bar graph indicates the ratio of Ki67 +cells to Sox2 +NSCs in the SGZ [WT, n = 4; *hGFAP-Cre;SmoM2*, n = 3]. Values represent mean ±SEM; ns: p>0.05, *p<0.05, **p<0.01. Student's *t*-test.

DOI: https://doi.org/10.7554/eLife.42918.005

The following source data and figure supplements are available for figure 2:

**Source data 1.** Extended numerical data and statistical analysis for *Figure 2—figure supplement 1*.
DOI: https://doi.org/10.7554/eLife.42918.006

**Figure supplement 1.** Excessive Shh signaling activity leads to abnormal morphology of DGs and increased cell death.
DOI: https://doi.org/10.7554/eLife.42918.004

**Figure supplement 1—source data 1.** Extended numerical data and statistical analysis for *Figure 2*.
DOI: https://doi.org/10.7554/eLife.42918.007

## Deletion of *Sufu* during DG development decreases NSC number and impairs adult neurogenesis

After development, NSCs in the DG are maintained until adulthood and produce neurons throughout life in the rodent brain. Given the widespread impairments in NSC numbers and proliferative capacity in the absence of Sufu, we next investigated the impact of these defects on adult neurogenesis. To label newborn neurons in adult mice, we administrated the thymidine analog 5-bromo-2'-deoxyuridine (BrdU) to 8 week old mice for 5 days and sacrificed the animal 3 days post-BrdU injection (*Figure 4A*). In hGFAP-Sufu-KO;Gli1^lacZ/+ mice, the number of BrdU +newborn neurons, as identified by DCX, was significantly decreased (*Figure 4B and C*). Accordingly, there was a significant reduction of total DCX + cells and Tbr2 +cells in hGFAP-Sufu-KO;Gli1^lacZ/+ mice compared with

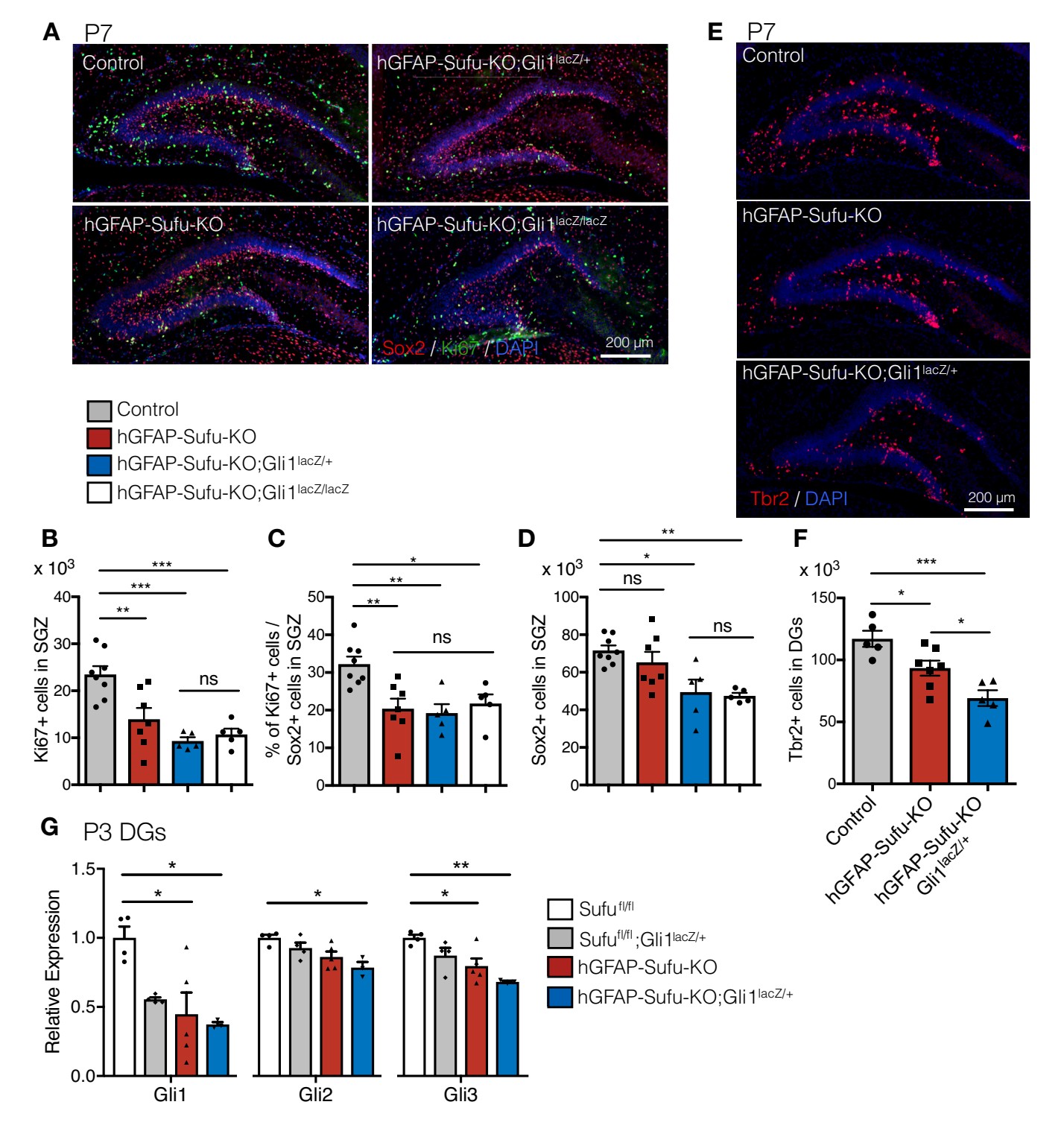

**Figure 3.** *Gli1* deletion increases the developmental defects in hGFAP-Cre Sufu[fl/fl] mice. (**A**) Representative immunofluorescence images of Sox2 (red), Ki67 (green) and DAPI (blue) in the DG of Control, hGFAP-Sufu-KO, hGFAP-Sufu-KO;Gli1[lacZ/+], and hGFAP-Sufu-KO;Gli1[lacZ/lacZ] mice at P7. (**B, C**) Quantification of Ki67 +cells (**B**) and ratio of Ki67 +proliferating cell population in Sox2 +cells (**C**) in SGZ [Control, n = 8; hGFAP-Sufu-KO, n = 7; hGFAP-Sufu-KO;Gli1[lacZ/+], n = 5; hGFAP-Sufu-KO;Gli1[lacZ/lacZ], n = 5]. (**D**) Sox2 +cells counts in SGZ from each group [Control, n = 8; hGFAP-Sufu-KO, n = 7; hGFAP-Sufu-KO;Gli1[lacZ/+], n = 5; hGFAP-Sufu-KO;Gli1[lacZ/lacZ], n = 5]. (**E**) Representative immunofluorescence images of Tbr2 (red) and DAPI (blue) in the DG of Control, hGFAP-Sufu-KO and hGFAP-Sufu-KO;Gli1[lacZ/+] mice at P7. (**F**) The number of Tbr2 +cells in DGs [Control, n = 7; hGFAP-Sufu-KO,

*Figure 3 continued on next page*

*Figure 3 continued*

n = 7; hGFAP-Sufu-KO;Gli1$^{lacZ/+}$, n = 5]. (G) qRT-PCR analyses of *Gli1*, *Gli2* and *Gli3* expression in the P3 DGs of *Sufu$^{fl/fl}$*, *Sufu$^{fl/fl}$;Gli1$^{lacZ/+}$*, hGFAP-Sufu-KO and hGFAP-Sufu-KO;Gli1$^{lacZ/+}$ mice [*Sufu$^{fl/fl}$*, n = 4; *Sufu$^{fl/fl}$;Gli1$^{lacZ/+}$*, n = 4, hGFAP-Sufu-KO, n = 5; hGFAP-Sufu-KO;Gli1$^{lacZ/+}$, n = 3]. Values represent mean ±SEM; ns: p>0.05, *p<0.05, **p<0.01, ***p<0.001. ANOVA with Tukey post-hoc tests.
DOI: https://doi.org/10.7554/eLife.42918.008

The following source data and figure supplements are available for figure 3:

**Source data 1.** Extended numerical data and statistical analysis for *Figure 3*.
DOI: https://doi.org/10.7554/eLife.42918.011

**Figure supplement 1.** Deletion of *Gli1* does not affect cell proliferation.
DOI: https://doi.org/10.7554/eLife.42918.009

**Figure supplement 1—source data 1.** Extended numerical data and statistical analysis for *Figure 3—figure supplement 1*.
DOI: https://doi.org/10.7554/eLife.42918.012

**Figure supplement 2.** Deletion of *Sufu* decreases proliferation of NSCs, but not neuronal precursors.
DOI: https://doi.org/10.7554/eLife.42918.010

**Figure supplement 2—source data 1.** Extended numerical data and statistical analysis for *Figure 3—figure supplement 2*.
DOI: https://doi.org/10.7554/eLife.42918.013

control mice (*Figure 4D* and *Figure 4—figure supplement 1A and B*). These data suggest that pre-natal deletion of *Sufu* reduced adult neurogenesis. To determine if impaired neurogenesis is due to decreased differentiation of NSCs into neurons, we next compared the ratio of DCX + newborn neurons to BrdU labeled cells. However, there was no difference between control, hGFAP-Sufu-KO and hGFAP-Sufu-KO;Gli1$^{lacZ/+}$ mice (*Figure 4E*), suggesting that the observed defects in adult neurogenesis is not because of failure of NSCs to undergo neuronal differentiation.

Impairments in adult neurogenesis may also arise from the failure of NSCs to undergo cell division. Therefore, we also examined the proliferation rate of adult NSCs by counting the proliferative Ki67 +population of Sox2 +cells. We found a comparable fraction of proliferating NSCs in the SGZ of DGs between control, hGFAP-Sufu-KO and hGFAP-Sufu-KO;Gli1$^{lacZ/+}$ mice (*Figure 4—figure supplement 1C and D*). This suggests that deletion of *Sufu* does not impair NSC proliferation at adult stages.

Adult NSCs are maintained in a quiescent state until stimulated to proliferate and produce neurons (*Encinas et al., 2011*; *Mira et al., 2010*). Since *Sufu* deletion reduced the number of Sox2 +cells in SGZ at P14, we investigated whether reduction of newborn neurons in hGFAP-Sufu-KO;Gli1$^{lacZ/+}$ mice was due to a reduction in the production or maintenance of the quiescent NSC pool. These quiescent NSCs display a radial morphology with fiber extending to the molecular layer (*Bignami and Dahl, 1974*; *Lugert et al., 2010*; *Rickmann et al., 1987*; *Sievers et al., 1992*). To determine the number of quiescent NSCs in adult DGs, we counted the number of Sox2 +cells in SGZ, which have a GFAP +radial fiber. The number of Sox2+/GFAP +radial NSCs was significantly reduced in both hGFAP-Sufu-KO and hGFAP-Sufu-KO;Gli1$^{lacZ/+}$ mice, with a greater reduction observed in hGFAP-Sufu-KO;Gli1$^{lacZ/+}$ mice (*Figure 4F and G*). Furthermore, we also calculated the ratio of DCX + cells to Sox2+/GFAP +radial NSCs to clarify the neurogenic competence of NSCs, and found no difference between control, hGFAP-Sufu-KO and hGFAP-Sufu-KO;Gli1$^{lacZ/+}$ mice (*Figure 4H*). This suggests that newborn neurons were produced from quiescent NSCs at comparable ratios and that the neurogenic competence was not impaired. Further, these findings supported our idea that decreased newborn neurons in hGFAP-Sufu-KO and hGFAP-Sufu-KO;Gli1$^{lacZ/+}$ mice results from reduction of NSC pool. These data suggest that deletion of *Sufu* in the developing DGs decreases the number of NSCs maintained through adulthood resulting in impaired adult neurogenesis.

## Deletion of *Sufu* impairs proliferation and expansion of NSCs in the DG at early postnatal stages

Proliferation of NSCs in the first postnatal week is critical for producing and maintaining NSCs until adult stages (*Youssef et al., 2018*). Elimination of proliferating cells in the first postnatal week, but not at 2–3 weeks, severely impairs the size of the NSC pool in adult DG. We have previously shown that long-lived NSCs of the DG are composed of Hh-responding cells in the ventral hippocampus at E17.5 (*Li et al., 2013*). Using Gli1$^{CreER/+}$::Rosa$^{Ai14/+}$mice treated with Tamoxifen at E17.5, we labeled

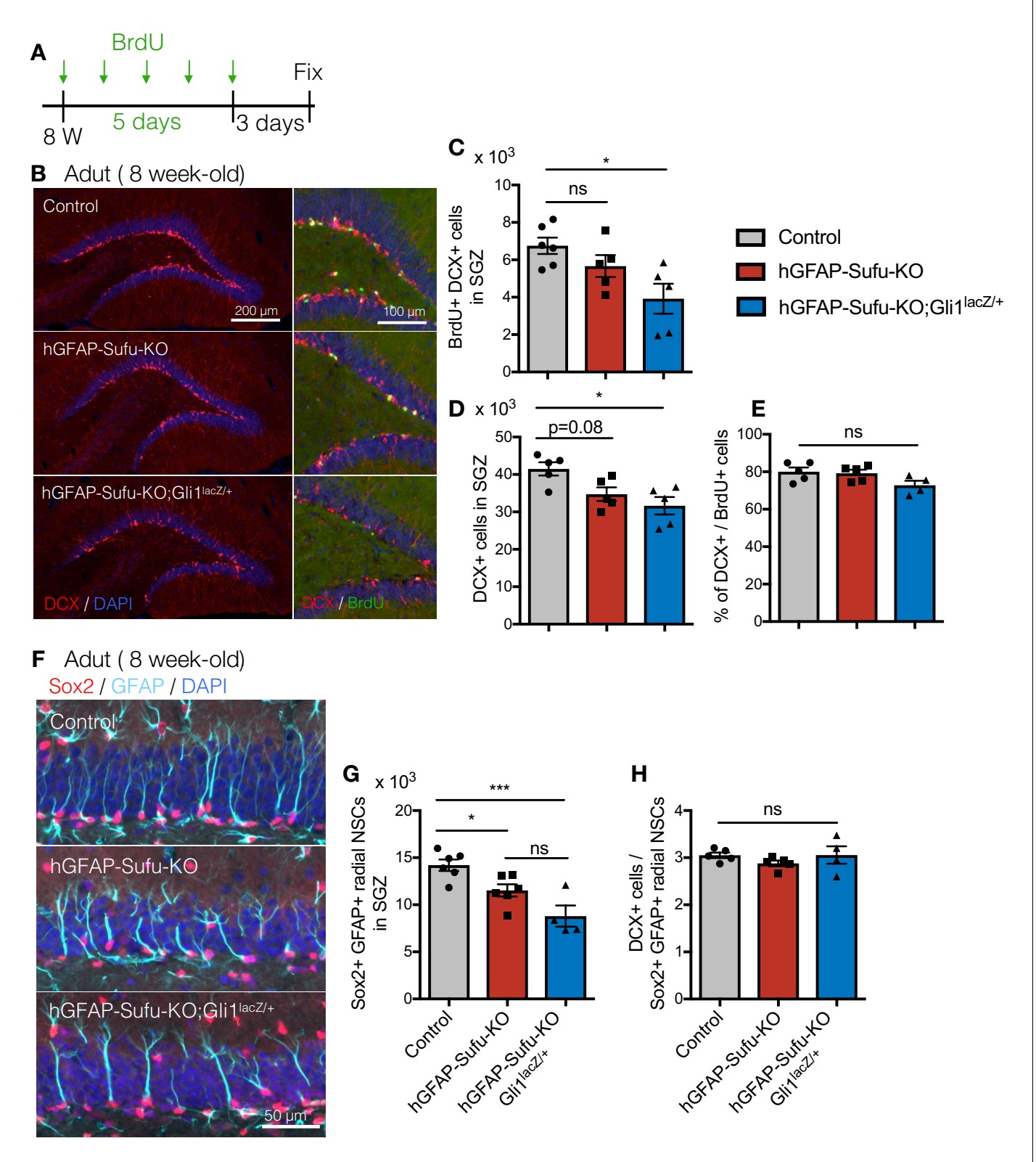

**Figure 4.** Deletion of *Sufu* during DG development decreases qNSCs pool in adult DGs. (**A**) Experimental scheme of BrdU injection. 8 week-old mice were injected with BrdU for 5 days and analyzed 3 days after last BrdU injection. (**B**) Representative immunofluorescence images for DCX (red) and DAPI (blue) in the DGs of control, hGFAP-Sufu-KO and hGFAP-Sufu-KO;Gli1$^{lacZ/+}$ mice. Magnified images of DCX (red), BrdU (green) and DAPI (blue) are shown to the right of each image. (**C, D**) Quantification of BrdU+/DCX + cells (**C**) and DCX + cells (**D**) in SGZ [Control, n = 6; hGFAP-Sufu-KO, n = 5;

*Figure 4 continued on next page*

*Figure 4 continued*

hGFAP-Sufu-KO;Gli1$^{lacZ/+}$, n = 5]. (**E**) The bar graph indicates the ratio of DCX + cells to BrdU +cells in SGZ [n = 5 biological replicates per group]. (**F**) Representative immunofluorescence images for Sox2 (red), GFAP (cyan) and DAPI (blue) in the SGZ of control, hGFAP-Sufu-KO and hGFAP-Sufu-KO; Gli1$^{lacZ/+}$ mice. (**G**) Quantification of Sox2+/GFAP +radial NSCs [Control, n = 6; hGFAP-Sufu-KO, n = 6; hGFAP-Sufu-KO;Gli1$^{lacZ/+}$, n = 4]. (**H**) The bar graph indicates the ratio of DCX + cells to Sox2+/GFAP +radial NSCs in SGZ [Control, n = 5; hGFAP-Sufu-KO, n = 5; hGFAP-Sufu-KO;Gli1$^{lacZ/+}$, n = 4]. Values represent mean ±SEM; ns: p>0.05, *p<0.05, ***p<0.001. ANOVA with Tukey post-hoc tests.

DOI: https://doi.org/10.7554/eLife.42918.014

The following source data and figure supplements are available for figure 4:

**Source data 1.** Extended numerical data and statistical analysis for *Figure 4*.
DOI: https://doi.org/10.7554/eLife.42918.016

**Figure supplement 1.** Deletion of *Sufu* during DG development decreases qNSCs pool in adult DGs.
DOI: https://doi.org/10.7554/eLife.42918.015

**Figure supplement 1—source data 1.** Extended numerical data and statistical analysis for *Figure 4—figure supplement 1*.
DOI: https://doi.org/10.7554/eLife.42918.017

Hh-responding cells and traced their migration during the first postnatal week. We found that Ai14 +cells were sparsely localized in the dorsal DG and fimbriodentate junction (FDJ) at P0 (*Figure 5A*). However, the number of Ai14 +cells dramatically increased with development and accumulated in the border between hilus and GCL from P3 to P7. The number of Sox2 +cells in Ai14 +cells of ventrally derived NSCs was significantly increased from P0 to P3, and P3-P7, respectively (*Figure 5B–D*). Using Gli1$^{CreER/+}$ mice, we also tested if inducing deletion of Sufu specifically in ventral NSCs decreases the number of NSCs. Gli1$^{CreER/+}$;Sufu$^{fl/fl}$;Ai14 mice showed a significant reduction in Sox2+/Ai14 +cells and Sox2+/Ki67+/Ai14 +cells at P7 (*Figure 5—figure supplement 1A–C*), which is consistent with the results from GFAP-Cre mice (*Figure 3*). Together, these findings suggest that the first postnatal week is a critical period for long-lived NSC expansion.

In hGFAP-Sufu-KO and hGFAP-Sufu-KO;Gli1$^{lacZ/+}$ mice, we found that the number of NSCs in adult DG was significantly reduced (*Figure 4F and G*). This might be attributed to impaired NSC expansion in the first postnatal week. To address this possibility, we labeled proliferating cells with BrdU at P0 or P3, and assessed the number of BrdU+/Sox2 +cells 2 hr post BrdU injection. At P0, there was no difference in the number of BrdU+/Sox2 +cells in both dorsal and ventral DGs between control, hGFAP-Sufu-KO and hGFAP-Sufu-KO;Gli1$^{lacZ/+}$ mice (*Figure 5E,F*). However, at P3, in dorsal, but not ventral DG, both hGFAP-Sufu-KO and hGFAP-Sufu-KO;Gli1$^{lacZ/+}$ mice showed significant reduction in BrdU+/Sox2 +cells compared with control (*Figure 5G and H*). These findings showed that Sox2 +NSCs in the dorsal DG remain proliferative during the expansion period of long-lived NSCs. Taken together, these data suggest that *Sufu* deletion impairs the proliferation of NSCs at a critical expansion period resulting in reduced number of quiescent NSCs in the adult DG.

## Deletion of *Sufu* leads to the premature transition of NSCs into quiescence during DG development

Our data show that *Sufu* deletion decreased NSC proliferation during the critical expansion period for long-lived NSCs, pointing to the likelihood that NSCs prematurely transitioned into a quiescent state. To test this, we utilized two thymidine analogs 5-Chloro-2-deoxyuridine (CldU) and 5-Iodo-2-deoxyuridine (IdU), and injected each thymidine analogs at different time points; CldU at P0, 3, 7 or 14 and IdU at 8 weeks old (*Figure 6A and B*). Because the thymidine analog is diluted as cells divide, cells that proliferated and stopped in the postnatal period will have detectable CldU, and therefore, when IdU is injected at adult stages, cells that became quiescent in developmental stages and then are reactivated in the adult will be double positive for CldU and IdU in adult stages. Double positive cells were observed more in mice injected with CldU at P3 and gradually decreased in groups injected with CldU at later postnatal stages (*Figure 6C and D*). Accordingly, the number of DCX+/CldU+/IdU +cells in newborn neurons was highest in groups injected at P3 (*Figure 6E and F*). Similarly, control mice injected with CldU at P3 had the highest number of CldU and IdU double positive cells at adult stages (*Figure 6G and H*). These data suggest that around P3-P7, NSCs in the control DG reduce their proliferation rate and become quiescent. However, in hGFAP-Sufu-KO and hGFAP-Sufu-KO;Gli1$^{lacZ/+}$ mice, the number of CldU and IdU double positive cells in CldU-injected groups at P3 or P7 was significantly decreased compared to control mice. Instead, detection of CldU and

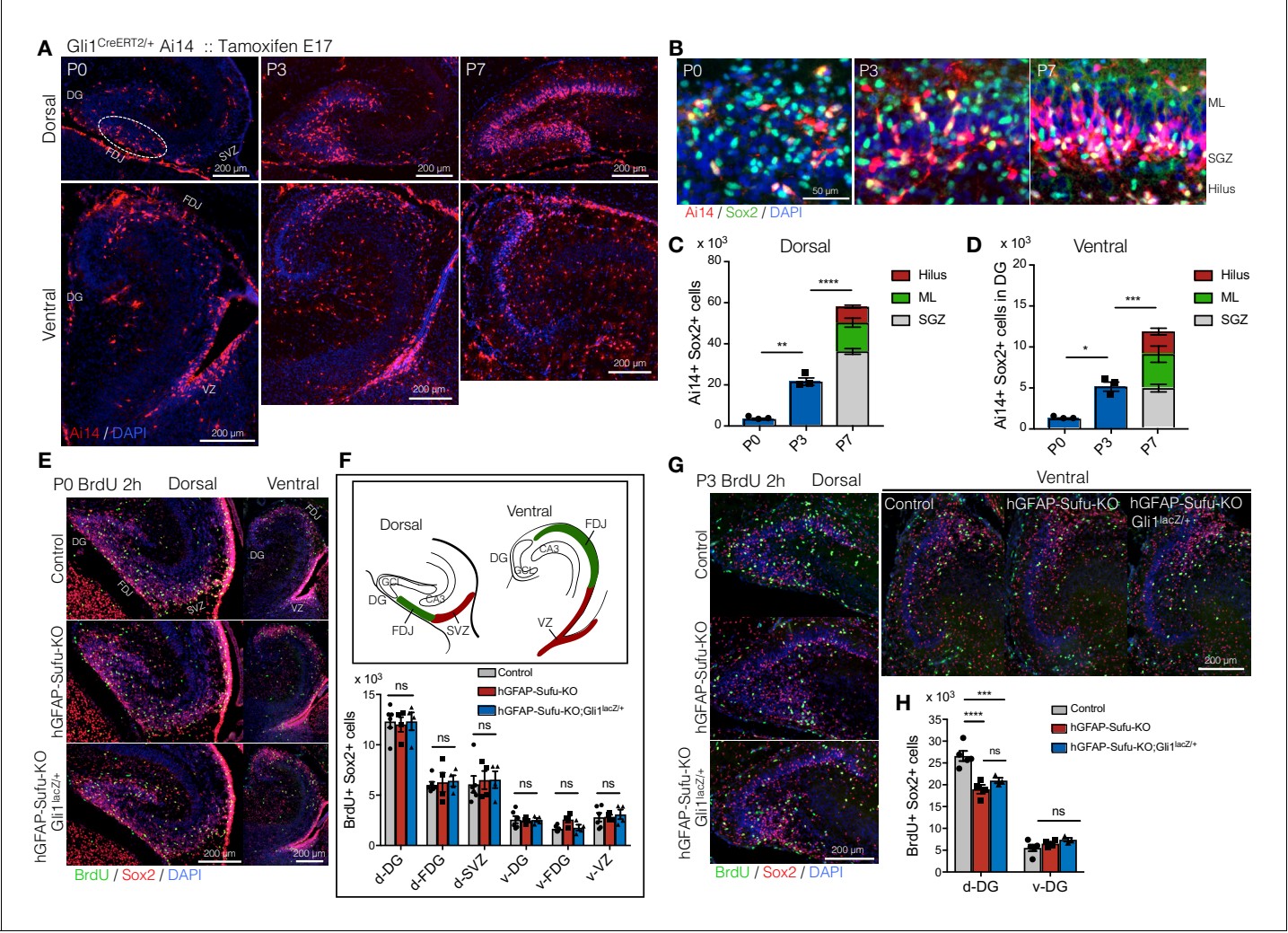

**Figure 5.** Long-lived NSCs expand in first postnatal week, and loss of Sufu impairs its expansion. (A) Fate tracing of Hh-responding cells at E17.5 in postnatal DG development. Gli1[CreERT2/+] Ai14 mice were treated with tamoxifen at E17.5 and Ai14 +Hh responding cells are analyzed in postnatal first week (P0-7). Representative immunofluorescence images for Ai14 (red) and DAPI (blue) in dorsal and ventral hippocampus in sagittal sections. Note that Hh-responding cells labeled by Ai14 at E17.5 sparsely appear in DGs at P0 and subsequently accumulate in SGZ. (B) Representative immunofluorescence images for Ai14 (red), Sox2 (green) and DAPI (blue) in SGZ. (C,D) Quantification of Ai14+/Sox2 +cells in dorsal (C) and ventral (D) DGs [n = 3 biological replicates per group]. The number of long-lived NSCs, indicated as Ai14+/Sox2 +cells, are increased in postnatal first week. (E) BrdU was injected at P0, and pups were sacrificed 2 hr later. Representative immunofluorescence images for BrdU (green), Sox2 (red) and DAPI (blue) in dorsal and ventral DGs of control, hGFAP-Sufu-KO and hGFAP-Sufu-KO;Gli1[lacZ/+] mice. (F) Schematic illustration of the dorsal and ventral DG at P0. d: dorsal, v: ventral, FDJ: fimbriodentate junction; SVZ: subventricular zone. The bar graph indicates the number of Sox2+/BrdU +cells in each region [Control, n = 6; hGFAP-Sufu-KO, n = 4; hGFAP-Sufu-KO;Gli1[lacZ/+], n = 4]. (G) BrdU was injected at P3, and pups were sacrificed 2 hr later. Representative immunofluorescence images for BrdU (green), Sox2 (red) and DAPI (blue) in dorsal and ventral DGs of control, hGFAP-Sufu-KO and hGFAP-Sufu-KO;Gli1[lacZ/+] mice. (H) Quantification of Sox2+/BrdU +cells in dorsal and ventral DGs [Control, n = 5; hGFAP-Sufu-KO, n = 4; hGFAP-Sufu-KO;Gli1[lacZ/+], n = 3]. Values represent mean ±SEM; ns: $p > 0.05$, *$p < 0.05$, **$p < 0.01$, ***$p < 0.001$, ****$p < 0.0001$. ANOVA with Tukey post-hoc tests.

DOI: https://doi.org/10.7554/eLife.42918.018

The following source data and figure supplements are available for figure 5:

**Source data 1.** Extended numerical data and statistical analysis for *Figure 5*.
DOI: https://doi.org/10.7554/eLife.42918.020

**Figure supplement 1.** Prenatal deletion of *Sufu* in NSCs of ventral hippocampus leads to reduction of NSCs in the postnatal DGs.
DOI: https://doi.org/10.7554/eLife.42918.019

**Figure supplement 1—source data 1.** Extended numerical data and statistical analysis for *Figure 5—figure supplement 1*.
DOI: https://doi.org/10.7554/eLife.42918.021

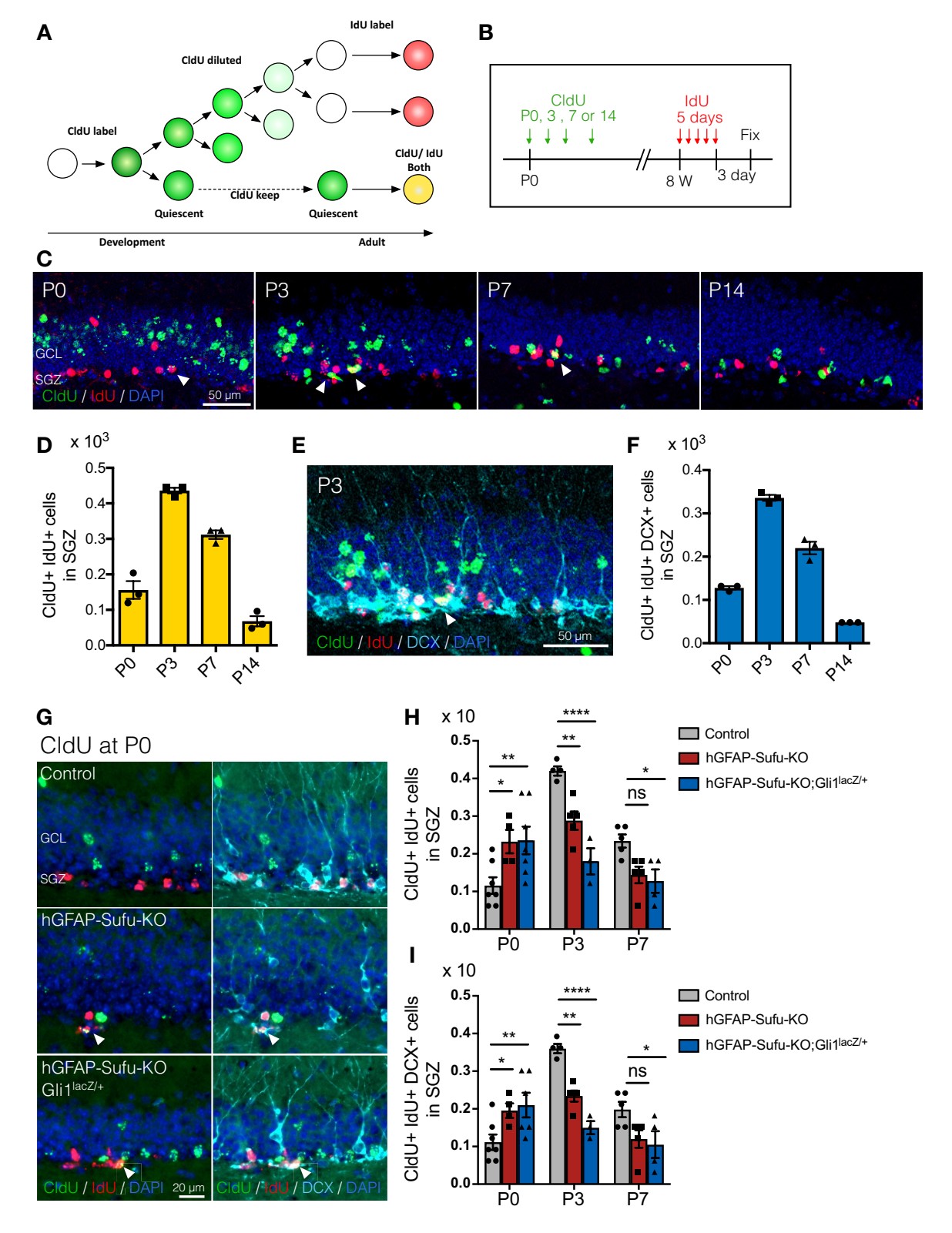

**Figure 6.** Deletion of *Sufu* prematurely induces the quiescent state transition. (**A**) Schematic illustration for labeling NSCs that established quiescent state during DG development using two thymidine analogs. CldU is diluted when cell divides. Thus CldU amount goes down if the cell continuously proliferates during DG development. However, if the cell becomes quiescent state, CldU amount is maintained until adult then the cell can be double-labeled with CldU and IdU if the CldU-labeled quiescent cell starts proliferation and incorporate IdU at adult. (**B**) Experimental scheme of CldU and IdU

*Figure 6 continued on next page*

Figure 6 continued

injection. CldU was injected at P0, P3, P7 or P14, and then IdU was injected for 5 days at 8 weeks old. The mice were sacrificed at 3 days after last IdU injection. (C) Representative immunofluorescence images for CldU (green), IdU (red) and DAPI (blue) in the SGZ of animals injected CldU at different stages. White arrowheads indicate the CldU/IdU double-labeled cells. (D) Quantification of CldU+/IdU +cells in the SGZ [n = 3 biological replicates per group]. (E) Representative immunofluorescence images for CldU (green), IdU (red), DCX (cyan) and DAPI (blue) in the SGZ of animal injected CldU at P3. White arrowheads indicate the CldU+/IdU+/DCX + cells. (F) Quantification of CldU+/IdU+/DCX + cells in the SGZ [n = 3 biological replicates per group]. (G) Representative immunofluorescence images for CldU (green), IdU (red), DCX (cyan) and DAPI (blue) in the SGZ of control, hGFAP-Sufu-KO and hGFAP-Sufu-KO;Gli1$^{lacZ/+}$ mice injected CldU at P0. White arrowheads indicate the CldU+/IdU+/DCX + cells. (H,I) Quantification of CldU+/IdU +cells (H) and CldU+/IdU+/DCX + cells (I) in the SGZ [P0 CldU: Control, n = 7; hGFAP-Sufu-KO, n = 4; hGFAP-Sufu-KO;Gli1$^{lacZ/+}$, n = 6; P3 CldU: Control, n = 4; hGFAP-Sufu-KO, n = 5; hGFAP-Sufu-KO;Gli1$^{lacZ/+}$, n = 3; P7 CldU: Control, n = 5; hGFAP-Sufu-KO, n = 5; hGFAP-Sufu-KO;Gli1$^{lacZ/+}$, n = 4]. Values represent mean ±SEM; ns: p>0.05, *p<0.05, **p<0.01, ****p<0.0001. ANOVA with Tukey post-hoc tests.

DOI: https://doi.org/10.7554/eLife.42918.022

The following source data and figure supplements are available for figure 6:

Source data 1 . Extended numerical data and statistical analysis for Figure 6.

DOI: https://doi.org/10.7554/eLife.42918.025

Figure supplement 1. Deletion of Sufu prematurely induces the quiescent state transition.

DOI: https://doi.org/10.7554/eLife.42918.023

Figure supplement 1—source data 1. Extended numerical data and statistical analysis for Figure 6—figure supplement 1.

DOI: https://doi.org/10.7554/eLife.42918.026

Figure supplement 2. Deletion of Sufu prematurely decreases the proliferation of NSCs during DG development.

DOI: https://doi.org/10.7554/eLife.42918.024

Figure supplement 2—source data 1. Extended numerical data and statistical analysis for Figure 6—figure supplement 2.

DOI: https://doi.org/10.7554/eLife.42918.027

IdU double positive cells were significantly increased at P0 compared to control. The number of DCX + cells double labeled with CldU and IdU was also decreased in P3 or P7 CldU injected groups, whereas it was increased significantly in P0 CldU injected groups (*Figure 6I*). These data suggest that NSCs in hGFAP-Sufu-KO and hGFAP-Sufu-KO;Gli1$^{lacZ/+}$ mice prematurely exited the proliferative state from P0-3 instead of at P3-P7. To support this idea, we further clarified the timing of the transition towards quiescence by investigating the number of CldU+/Sox2+/GFAP +radial NSCs generated at each time point. Similar to our findings from the CldU and IdU double labeling experiments above, we found that the number of Sox2+/GFAP +radial NSCs labeled with CldU was decreased significantly in P3 and P7 CldU injected groups, whereas it was increased significantly in P0 CldU injected groups in hGFAP-Sufu-KO and hGFAP-Sufu-KO;Gli1$^{lacZ/+}$ mice (*Figure 6—figure supplement 1A and B*). In line with these observations, we also found that the Ki67 +proliferating population in Sox2 +NSCs was significantly reduced at P3, but not at P0 in hGFAP-Sufu-KO and hGFAP-Sufu-KO;Gli1$^{lacZ/+}$ mice compared with control (*Figure 6—figure supplement 2A–D*). Comparable proportions of proliferating NSCs were found between three groups at P0 in the DG and FDJ where ventral-derived NSCs were localized at this time point (*Figure 6—figure supplement 2A and B*). However, we found that actively proliferating Sox2 +cells was significantly decreased in the SGZ of hGFAP-Sufu-KO and hGFAP-Sufu-KO;Gli1$^{lacZ/+}$ mice at P3 and P7 (*Figure 6—figure supplement 2C and D*). Taken together, these data suggest that deletion of Sufu precociously reduces proliferation of NSCs and leads to premature transition to quiescent state and thus a smaller pool of quiescent NSCs in the adult dentate.

## Discussion

Quiescence is key to maintaining the NSC pool and is critical for enabling lifelong neurogenesis, so understanding how actively dividing NSCs become quiescent during development is vitally important. Here, we demonstrated that long-lived NSCs dramatically expand in the first postnatal week before entering the quiescent state over several days. Sufu deletion impairs the ability of long-lived NSCs to expand in the first postnatal week, which results in the premature entry of NSCs into the quiescent state (*Figure 7*). This defect is a result of decreased Shh signaling activity as a result of Sufu deletion in NSCs.

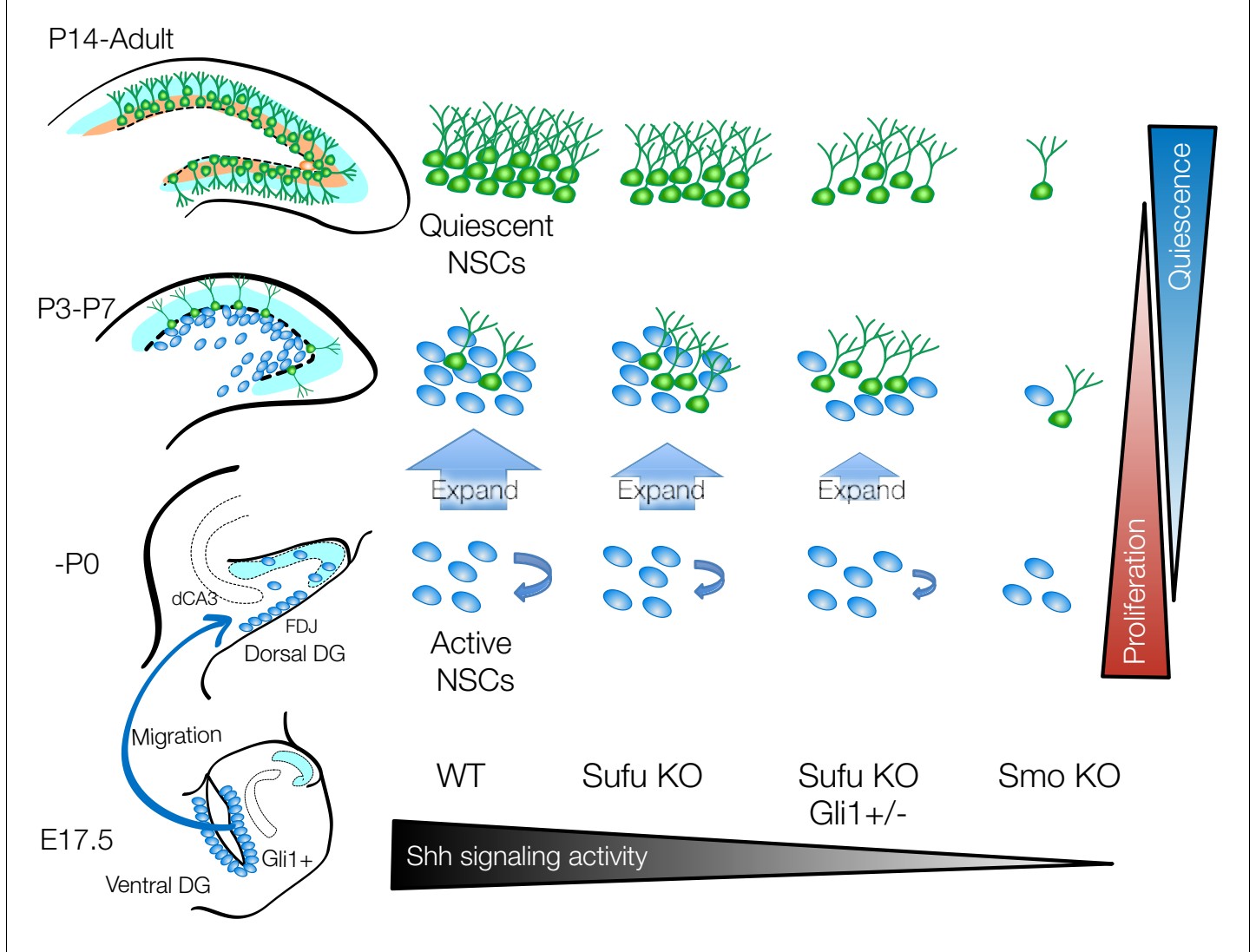

**Figure 7.** Sufu is important for the perinatal expansion and quiescent state transition of dentate NSCs. Schematic summary illustrating the role of Sufu and Shh signaling activity in initial production, expansion and quiescent state transition of NSCs during DG development. Long-lived NSCs are produced from Hh-responding cells in ventral hippocampus by Shh stimuli at E17.5 and migrate to dorsal and ventral DGs. This subset of NSCs expands in first postnatal week and subsequently become quiescent state. Sufu controls shh signaling activity in NSCs during DG development. *Sufu* deletion decreases the Shh signaling activity and leads to impaired expansion of long-lived NSC, resulting in premature quiescent state transition and small NSC pool in adult. On the other hand, complete ablation of Shh signaling activity by deleting *Smo* impairs initial production of long-lived NSCs at the beginning of DG development and compromises the establishment of neurogenic niche.

DOI: https://doi.org/10.7554/eLife.42918.028

Surprisingly, our data showed that deletion of *Sufu* in GFAP-expressing NSCs decreased Shh signaling activity. Gli1-LacZ + cells were dramatically reduced in the embryonic ventral hippocampus and were abolished in the postnatal DG. Shh signaling is crucial for the initial production of dentate NSCs in the embryonic stage. *Smo* deletion in NSCs severely impairs initial NSC production and leads to decreased size of GCL (*Han et al., 2008*; *Li et al., 2013*). Although *Sufu* deletion impaired proliferation of NSCs in the first postnatal week, we still observed comparable number of proliferating NSCs at P0, indicating that initial NSC production at embryonic stages was not affected. Whether this process is dependent on Shh signaling is yet to be investigated, since we cannot exclude the possibility that very low levels of Shh signaling activity, likely undetectable by LacZ activity, are sufficient to produce the proper number of NSCs in embryonic stages.

Sufu is primarily known for its role as a negative regulator of Shh signaling pathway through several mechanisms - promoting the formation of Gli repressors, removing Gli1 from nucleus, and recruiting transcription repressors to Gli-target genes sites (*Barnfield et al., 2005*; *Cheng and Bishop, 2002*; *Kise et al., 2009*; *Kogerman et al., 1999*). However, recent studies provide evidence that Sufu may act as both a negative and positive regulator of Shh signaling under specific conditions. For example, in mouse embryonic fibroblast (MEF), Shh ligand binding triggers Sufu translocation to the nucleus with Gli1, where it binds to and facilitates cytoplasmic export of Gli3 repressor, thereby enhancing Shh signaling activity (*Zhang et al., 2017*). Furthermore, increasing the amount of Sufu added into $Sufu^{-/-}$ MEF compromises Hh-responsiveness in the absence of exogenous Shh, which is consistent with negative function of Sufu for Shh signaling. However, in the presence of exogenous Shh, Hh-responsiveness of $Sufu^{-/-}$ MEF is dramatically elevated with the addition of increasing levels of Sufu, an effect that was not observed when Ptch1 was added (*Chen et al., 2009*). These reports demonstrate that Sufu can act to enhance or maximize Shh signaling activity. Considering these roles, decreased Shh signaling activity in Sufu KO mice could result from the failure to maximize Shh signaling activity through mechanisms that likely involve stabilization of Gli transcription factors. Supporting this, our data showed that *Gli1* and *Gli2* expression were downregulated in hGFAP-Sufu-KO;Gli1$^{lacZ/+}$ mice in which expression of Shh signaling target genes were significantly reduced. Altogether, our data demonstrate multiple regulatory roles of Sufu in Shh signaling pathway that is dependent on cell type and context.

We found that deletion of *Sufu* increased the dependency of proper DG development on Gli1 function. Gli1 is not necessary for initial activation of Shh signaling (*Bai et al., 2002*; *Park et al., 2000*). Therefore, *Gli1* deletion normally does not cause any significant developmental defects (*Park et al., 2000*). However, the phenotypes in hGFAP-Sufu-KO mice clearly worsen when combined with *Gli1* deletion, suggesting that Gli1 function becomes necessary in the absence of Sufu. Previous studies show that Sufu is important for the stabilization of Gli2 and Gli3 proteins (*Makino et al., 2015*; *Wang et al., 2010*). Loss of Sufu results in diminishing Gli2 full-length activator. Sufu competitively binds to Gli2 and Gli3 with speckle-type POZ protein (Spop), which recruits ubiquitin ligases and degrades the full-length forms of Gli2 and Gli3 (*Chen et al., 2009*; *Wang et al., 2010*; *Wen et al., 2010*). These finding indicates that loss of Gli2 activators in the absence of Sufu increases the requirement for Gli1 function to maintain Shh signaling activity. Indeed, Gli1 is able to compensate for lost Gli2 activator function and rescue the developmental defects of Gli2 knockout mice (*Bai and Joyner, 2001*).

Expansion of long-lived NSCs must occur in the first postnatal week during which time ventral-derived NSCs dramatically increase their numbers in both the ventral and dorsal DG. Interestingly, *Sufu* deletion only impaired the proliferation of NSCs in the dorsal DG, but not ventral DG. This difference could be due to underlying molecular differences between NSCs and the surrounding cells residing in these regions. Lineage tracing of Shh expressing cell shows that neurons in the medial entorhinal cortex and hilar mossy cells function as the local source of Shh ligand for ventral derived NSCs, while hair mossy cells in the dorsal DG, but not ventral DG, are the source of Shh (*Li et al., 2013*). Conditionally removing Shh ligand from these local neurons abolishes signaling in Hh-responding cells and results in reduction of NSCs number, indicating that local Shh ligands are important for the activation of Shh signaling in NSCs after migration to the dorsal DGs (*Li et al., 2013*). Similar to mice lacking Shh, our data showed that deletion of *Sufu* abolished Hh-responding cells in postnatal DGs, followed by the reduction of proliferating NSCs in dorsal DGs. These findings indicate that dorsal and ventral NSCs use distinct developmental approaches to navigate the transition to produce long-lived NSCs. Dorsal NSCs must rely on Sufu to ensure optimal Shh signaling activity suitable for the expansion of long-lived NSCs during DG development, while NSCs in the ventral DG appear able to expand and become quiescent without Sufu.

Our data suggest that Shh signaling activity must be continuously maintained to promote NSC expansion and that eventual reduction in Shh signaling activity promotes NSCs to transition into the quiescent state. Supporting this, we found that Gli1-LacZ + cells were abundant in the ventral ventricular zone but progressively decreased after migration into the DG in the first postnatal week. Additionally, we found that constitutive activation of Shh signaling in NSCs, by conditional expression of SmoM2, increased proliferating NSCs and prevented transition into quiescence. These observations suggest that Sufu is important in sustaining NSC proliferation until NSCs begin to transition into a quiescent state. However, we also observed in SmoM2 mutants that subpopulations of NSCs

become quiescent, indicating that reduction of Shh signaling activity alone is not sufficient to initiate quiescence and that other signaling mechanisms are involved in this process. Indeed, several extra-cellular factors can regulate the quiescent state of NSCs, such as Bone Morphogenetic Proteins (BMPs), Notch, and gamma-aminobutyric acid (GABA) (*Kawaguchi et al., 2013*; *Mira et al., 2010*; *Song et al., 2012*). These molecules are secreted by granule neurons, astrocytes, microglia, and interneurons in the DG and are likely sources of signals for migrating ventral-derived NSCs (*Bonaguidi et al., 2011*; *Bond et al., 2014*; *Kawaguchi et al., 2013*; *Mira et al., 2010*; *Song et al., 2012*; *Yousef et al., 2015*). The activity of these signaling pathways, and the simultaneous reduction in Shh signaling with development, may coordinately function to ensure successful transition of NSCs into the quiescent state during DG development.

# Materials and methods

## Key resources table

| Reagent type (species) or resource | Designation | Source or reference | Identifiers | Additional information |
|---|---|---|---|---|
| Genetic reagent (*M. musculus*) | Sufu<sup>flox/flox</sup> | PMID: 20074523 | RRID: MGI:4840420 | Dr. Chi-Chung Hui (University of Toronto) |
| Genetic reagent (*M. musculus*) | hGFAP-Cre | PMID: 11668683 | RRID: MGI:2179048 | Jackson Laboratory (Stock:004600) |
| Genetic reagent (*M. musculus*) | Gli1<sup>CreERT2/+</sup> | PMID: 15315762 | RRID: MGI:3053957 | Jackson Laboratory (Stock:007913) |
| Genetic reagent (*M. musculus*) | Gli1<sup>LacZ/+</sup> | PMID: 12361967 | RRID: MGI:J:79392 | Jackson Laboratory (Stock:008211) |
| Genetic reagent (*M. musculus*) | Rosa-Ai14 | PMID: 20023653 | RRID: MGI:J:155793 | Jackson Laboratory (Stock:007908) |
| Genetic reagent (*M. musculus*) | SmoM2-YFP | PMID: 15107405 | RRID: MGI:3576373 | Jackson Laboratory (Stock:005130) |
| Antibody | Mouse monoclonal anti-Ki67 | BD Biosciences | RRID: AB_396287 | IHC (1:500) |
| Antibody | rabbit monoclonal anti-Sox2 | Abcam | RRID: AB_10585428 | IHC (1:250) |
| Antibody | Rabbit polyclonal anti-Tbr2 | Abcom | RRID: AB_778267 | IHC(1:250) |
| Antibody | rabbit polyclonal anti-DCX | Abcam | RRID: AB_732011 | IHC (1:1000) |
| Antibody | rat monoclonal anti-GFAP | Zymed | RRID: AB_2532994 | IHC (1:500) |
| Antibody | chickin polyclonal anti-GFAP | Millipore | RRID: AB_177521 | IHC (1:1000) |
| Antibody | rat monoclonal anti-RFP | Chromotek | RRID: AB_2336064 | IHC (1:1000) |
| Antibody | rat monoclonal anti-BrdU | Abcam | RRID: AB_305426 | IHC (1:500) CldU detection |
| Antibody | mouse monoclonalanti-BrdU | BD Biosciences | RRID: AB_400326 | IHC (1:100) IdU detection |
| Antibody | rabbit monoclonalaniti-Cleaved Caspase 3 | Cell Signaling | RRID: AB_2070042 | IHC (1:250) |
| Software | ImageJ | NIH | RRID: SCR_003070 | Cell counting |
| Software | Prism 7 | Graphpad | RRID: SCR_002798 | Statistuc analysis |

## Animals

Mice carrying the floxed Sufu allele (Sufu^fl, 'RRID:MGI:4840420') were kindly provided by Dr. Chi-Chung Hui (University of Toronto) and were genotyped as described elsewhere (*Pospisilik et al., 2010*). The following mouse lines were obtained from Jackson Laboratory (Bar Harbor, Maine): Gli1-CreERT2/+ (stock #007913, 'RRID:MGI:3053957'), Gli1^LacZ/+ (Stock #008211, 'RRID:MGI:J:79392'), Rosa-AI14 (Stock #007908, 'RRID:MGI:J:155793'), SmoM2 (Stock #005130, 'RRID:MGI:3576373'), hGFAP-Cre (Stock #004600, 'RRID:MGI:2179048'). Both male and female mice were analyzed with no distinction. All mice used in this study were maintained on a 12 hr light/dark cycle with free access to food and water. The day of vaginal plug was considered embryonic day 0.5. Mouse colonies were maintained at University of California San Francisco (UCSF) in accordance with National Institutes of Health and UCSF guidelines.

## Tamoxifen and thymidine analog administration

Tamoxifen (Sigma) was dissolved in corn oil at 10 mg/ml. Pregnant mice were intraperitoneally administered 2 mg of tamoxifen with 27-gauge needles. For 5-Bromo-2'-deoxyuridine (BrdU) labeling, mice were subcutaneously (neonatal pups) or intraperitoneally (adult mice) injected with BrdU (Sigma) dissolved in saline (0.9% NaCl) at a dose of 50 mg/kg. For two thymidine analog labeling, 5-Chloro-2'-deoxyuridine (CldU, Sigma) dissolved in saline was subcutaneously injected into neonatal pups at a dose of 42.5 mg/kg followed by a single intraperitoneal injection of 57.5 mg/kg 5-Iodo-2'-deoxyuridine (IdU, Sigma) by for five days at 8 weeks old. IdU was dissolved in 0.015N NaOH and neutralized to pH 7.0 with 2N HCl before injection.

## Tissue preparation

To prepare embryonic brain tissue, pregnant mice were sacrificed on the indicated developmental day, and embryos were perfused successively with phosphate-buffered saline (PBS) and ice-cold 4% paraformaldehyde (PFA) in PBS, pH 7.2. For preparation of postnatal and adult brains, pups and adult mice were deeply anesthetized before perfusion with 4% PFA in PBS. Brains were dissected and postfixed with 4% PFA in PBS overnight at 4°C. For cryoprotection, fixed brains were stored in 30% sucrose in PBS at 4°C. The brain was embedded in optimal cutting temperature (OCT) compound (Tissue Tek, Sakura Finetek, 25608–930) and frozen at −80°C for cryosectioning. Frozen brains were serially sectioned with Leica CM 1850 (Leica Microsystems, Wetzlar, Germany) in the coronal or sagittal plane at 16 μm thickness. Every fifteenth sections were serially mounted on individual Colorfrost Plus Microscope Slides (Fisher Scientific) in order from anterior to posterior (coronal section) or medial to lateral (sagittal section), and preserved at −20°C until use.

## LacZ staining and in situ hybridization

Animals for LacZ staining were perfused with 4% paraformaldehyde (PFA) and the dissected brains were postfixed with 4% PFA for 2 hr at 4°C. Cryosections were washed with PBS, and X-gal staining was developed at 37°C overnight in the staining solution (5 mM $K_3Fe(CN)_6$, 5 mM $K_4Fe(CN)_6$, 5 mM EGTA, 0.01% deoxycholate, 0.02% NP40, 2 mM $MgC1_2$, and 1 mg/ml X-gal). Sections were postfixed with 10% formalin at room temperature overnight, followed by counterstain with nuclear-fast red (H-3403, Vector Laboratories) at room temperature for 10 min before proceeding for dehydration (70%, 95%, 100% ethanol, xylene twice) and coverslipping with Mount-Quick (Ted Pella).

## Immunohistochemistry

Cryosections were washed with PBS and blocked for 1 hr at room temperature with blocking solution (10% Lamb serum and 0.3% Triton X-100), and incubated overnight at 4°C with primary antibodies diluted in blocking solution. The following primary antibodies were used in this study: mouse anti-Ki67 (1:500; BD Biosciences, 550609, RRID:AB_396287); rabbit anti-Sox2 (1:1000; Abcam, ab92494, RRID:AB_10585428); rabbit anti-Tbr2 (1:1000; Abcam, ab23345, RRID:AB_778267); rabbit anti-DCX (1:1000; Abcam, ab18723, RRID:AB_732011); rat anti-GFAP (1:500; Zymed, 13–300, RRID:AB_2532994); chickin anti-GFAP (1:1000; Millipore, RRID:AB_177521); rat anti-RFP (1:1000, Chromotek, 5f8-100, RRID:AB_2336064); rat anti-BrdU (for BrdU or CldU detection) (1:500, Abcam, ab6326, RRID:AB_305426) and mouse anti-BrdU (for IdU detection)(1:100, BD Biosciences, 347580, RRID:AB_400326); rabbit anti-cleaved caspase 3 (1:250; Cell signaling #9661, RRID:AB_2070042). For staining

of Ki67, Tbr2, Sox2 and thymidine analogs, sections were heated in 10 mM Citric acid pH 6.0 on boiling water bath for 10 min prior to blocking. For staining of Ai14 together with Ki67 and Sox2, sections were heated in 10 mM Citric acid pH 6.0 at 70 °C for 3 hr prior to blocking. After three washes in PBS, sections were incubated for 2 hr with corresponding secondary antibodies. After a final rinse with PBS, sections were mounted on glass slides with Prolong gold antifade reagent (Thermo Fisher Scientific).

## Cell counting

Images were acquired using a Zeiss LSM 780 confocal microscope or Axio Scan Z.1 (Carl Zeiss). NIH ImageJ (RRID:SCR_003070) was used to count the cells. Brain sections were serially mounted on individual fifteen slides glasses. Cell counting was performed on every fifteenth sections containing DG at the same anatomical level between each group, and marker-positive cells were counted in the series of collected sections throughout the indicated areas in the DG. The total number of marker-positive cells in each indicated areas was obtained by multiplying the resultant counts by 15 (according to the interval between sections).

## Real-time PCR

Total RNA was isolated using TRIzol Reagent (Thermo Fisher Scientific) and RNeasy Mini Kit (QIA-GEN), according to the manufacturer's instructions, and each sample was reverse-transcribed using a SuperScript IV cDNA Synthesis Kit (Invitrogen). Quantitative PCR reactions were performed using a SYBR Select Master Mix for CFX, and transcript expression was measured via CFX384 Touch Real-Time PCR Detection System (Bio-Rad). Expression levels of each gene were normalized to RNA polymerase II subunit A (polr2a) and calculated relative to the control. The primers used for this study are listed in *Supplementary file 1*.

## Statistical analysis

At least three mice per group were analyzed. Statistical analyses were performed using either Student's t-test or Welch's *t*-test (for comparisons between two groups); one-way or two-way ANOVA with Tukey's multiple comparison test (for multiple groups comparison) with Prism software (Graphpad, RRID:SCR_002798). All experiments were independently replicated at least three times. Differences were considered statistically significant at $p < 0.05$. Asterisks indicate significant differences (*<0.05; **<0.01, ***<0.001, ****<0.0001).

# Acknowledgements

We thank members of the SJP lab for helpful discussions, in particular OR Yabut and B Tran for technical help, suggestions, and helping to write this manuscript. This research was supported by NIH grant R01 NS075188 (SJP), Sasakawa Scientific Research Grant (HN), The Uehara Memorial Foundation (HN), JSPS Overseas Research Fellowships (HN).

# Additional information

### Funding

| Funder | Grant reference number | Author |
| --- | --- | --- |
| Uehara Memorial Foundation | | Hirofumi Noguchi |
| Japan Society for the Promotion of Science | Overseas Research Fellowship | Hirofumi Noguchi |
| Japan Science Society | Sasakawa Scientific Research Grant | Hirofumi Noguchi |
| National Institutes of Health | R01 NS075188 | Samuel J Pleasure |

The funders had no role in study design, data collection and interpretation, or the decision to submit the work for publication.

## Author contributions
Hirofumi Noguchi, Conceptualization, Resources, Supervision, Funding acquisition, Investigation, Visualization, Methodology, Writing—original draft, Project administration, Writing—review and editing; Jesse Garcia Castillo, Conceptualization, Formal analysis, Validation, Investigation, Methodology, Writing—original draft, Writing—review and editing; Kinichi Nakashima, Formal analysis, Validation, Investigation; Samuel J Pleasure, Data curation, Supervision, Writing—review and editing

## Author ORCIDs
Hirofumi Noguchi (ID) http://orcid.org/0000-0002-9779-4956
Samuel J Pleasure (ID) http://orcid.org/0000-0001-8599-1613

## Ethics
Animal experimentation: Mouse colonies were maintained at University of California San Francisco (UCSF) in accordance with National Institutes of Health and UCSF guidelines. Animal studies were approved by the Institutional Animal Care and Use Committee of UCSF (Protocol # AN176415-01A and AN165562-02A).

## Decision letter and Author response
Decision letter https://doi.org/10.7554/eLife.42918.033
Author response https://doi.org/10.7554/eLife.42918.034

# Additional files

## Supplementary files
• Supplementary file 1. The primers for qPCR analysis.
DOI: https://doi.org/10.7554/eLife.42918.029
• Transparent reporting form
DOI: https://doi.org/10.7554/eLife.42918.030

## Data availability
All data generated or analyzed in this study are included in the manuscript and supporting files.

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
