## [Decision Letter]

Thank you for submitting your article "Suppressor of Fused controls perinatal expansion and quiescence of future dentate adult neural stem cells" for consideration by *eLife*. Your article has been reviewed by three peer reviewers, one of whom, Francois Guillemot, is a member of our Board of Reviewing Editors, and the evaluation has been overseen by Jonathan Cooper as the Senior Editor.

The reviewers have discussed the reviews with one another and the Reviewing Editor has drafted this decision to help you prepare a revised submission.

This is a timely manuscript that examines the mechanisms underpinning the expansion and entry into quiescence of dentate neural stem cells at early postnatal stages. The data supports a role of Sonic hedgehog (Shh) signaling regulated by Suppressor of Fused (Sufu) in this process. There is much interest in understanding how adult neural stem cells are generated during brain development and this manuscript has the potential to make a significant contribution in this field.

Summary:

The authors have previously shown that neural stem cells (NSCs) from the ventral hippocampus seed the subgranular zone of the postnatal dentate gyrus to form the adult NSC niche – a process that requires Shh signaling. This new work examines the effect of Sufu deletion on the establishment of the NSC pool in the mouse hippocampus.

Sufu is usually considered a repressor of the Shh pathway although it has also been shown to stimulate Shh signaling in certain contexts. This is the case here where the phenotypes arising from *Sufu* deletion resemble the phenotypes of a Smo floxed mutant mouse they analysed in their previous work (Li et al., 2013).

The authors conclude from a number of CldU/IdU incorporation and immunohistochemistry experiments that in wildtype animals, many NSCs enter into quiescence at the end of the first postnatal week. This is later than in the other adult NSC niche where NSCs lining the lateral ventricles enter quiescence at embryonic stages. They also find that decreasing Shh signaling by several mechanisms including *Sufu* deletion leads to precocious entry into quiescence, fewer NSCs in the hippocampus at 8 weeks of age and reduced neurogenesis at later time points.

The most novel aspect of this study is that it defines the precise timing of proliferation and entry into quiescence of NSCs during DG development and implicates downregulation of Shh signaling in promoting the transition of NSCs from proliferation to quiescence during development of the DG and thereby in determining the size of the adult NSC population.

Less novel is the establishment of a role of Sufu in enhancing Shh signaling, as Sufu is known to either stimulate or suppress Shh signaling depending on the cellular context. The implication of Shh signaling in the generation and proliferation of NSCs in the DG at early postnatal stages is also not novel as this was already shown in their previous paper (Li et al., 2013).

Overall, the data is well presented, the figures are of excellent quality and the paper well written, and the data mostly supports the conclusions reached (although not entirely, see below). However, there are several issues which will need to be resolved prior to publication.

Essential revisions:

1) The authors show that there is a large expansion of NSCs during normal development of the dentate gyrus between P0 to P7 (Figure 5A-D) and they claim that deletion of *Sufu* causes NSCs to precociously transition to a quiescent state (P0-3 instead of P3-7 normally), thus reducing the pool of quiescent NSCs in the adult (Figure 4F). However, Figure 2B clearly shows that there is no difference in the number of Sox2+ NSCs in the dentate gyrus of Sufu KO mice at P7. If the expansion of NSCs occurs from P0 to P7, then the decrease in NSC number should be apparent by P7. In agreement with this, the model presented in Figure 7 suggests that there are fewer NSCs (quiescent and dividing combined) in the Sufu KO at P3-7, but the data in Figure 2B clearly shows that there is no difference in the number of Sox2+ NSCs at P7. Along the same line, if there is an excessive proliferation of NSCs in SmoM2 mice (Figure 2G, H), why is their number unchanged at P7 (Figure 2F)? The authors should comment on these apparent contradictions and may need to amend their model.

2) The authors show in Figure 5G, H that Sufu KO causes reduced proliferation of NSCs in the dorsal but not ventral dentate gyrus at P3. However, in Figure 1C it appears as though most of the Shh-responsive cells are in the posterior/ventral dentate gyrus at P7, which is presumably true for P3 as well. Can the authors explain this discrepancy?

3) The authors show in Supplementary Figure 1 that Gli1 deletion has no effect on the number or proliferation of NSCs at P7 in the dentate gyrus and conclude that "Gli1 is largely dispensable [for forebrain development]". If this is true, then can the authors comment on why they think that a Gli1-lacZ reporter line is a good reporter of the level of Shh signaling required for development of the dentate gyrus (Figure 1)? In light of the fact that Sufu KO unexpectedly decreases Gli1-lacZ reporter expression, is there another way to confirm that Sufu KO decreases Shh signaling in the developing dentate gyrus? Related to this, the interpretation of the data on the regulation and function of Gli genes in wildtype and Sufu KO mice in Figure 3G is somewhat confusing and need to be clarified (subsection “In the absence of Sufu, Gli1 function becomes responsible for proper proliferation of NSCs during DG development”).

4) Important for the interpretation of the data is the assumption that DG development is not affected until the first postnatal week. The authors show that total Sox2+ cell numbers are unchanged. However, this appears to be a rather superficial read out and should be expanded by other measures (e.g., estimates of total neuronal and glial populations). If *Sufu* deletion affects the formation of the DG, all effects could be at least be partially explained by secondary effects. Using a conditional Cre line to test later stages of *Sufu* deletion (e.g., early postnatal) would be one way to address this concern. Related to this, the study focuses on the proliferation and quiescence of NSCs, yet much of the analysis is done by quantifying Sox2+ cells, which include both NSCs and the larger population of IPCs. Shh/Sufu might regulate the proliferation of IPCs as well as NSCs and it is unclear how much the changes in cell proliferation observed in loss and gain of function mutants concern specifically NSCs. The authors should use a more specific NSC marker or reassess their conclusions.

5) In Figure 4, the authors analysed neurogenesis in the adult DG and found reduced numbers of radial glia-like NSCs and reduced rate of neuron production, however it is unclear whether NSC proliferation is altered (percentage of NSCs that are Ki67+) and whether the total number of DCX+ cells produced decreases in adult Sufu KO mice.

6) The authors overinterpret their data regarding the role of Sufu in regulation of Shh signaling and NSC quiescence. The last sentence of the Introduction states that "Sufu (…) is important for ensuring (…) the timely transition to a quiescent state during DG development". However there is no evidence that Sufu is implicated in determining the timing of the transition to quiescence. The authors add at the beginning of the Discussion: "Thus Sufu modulates the timing of quiescence of NSCs in DG development by controlling NSCs expansion via modulation of Shh signaling activity". Again the paper does not provide evidence that Sufu normally modulates Shh activity and regulates the timing of quiescence of NSCs. Changes in Shh signaling and in timing of NSC quiescence are observed in Sufu mutant mice but there is no evidence that entry of NSCs in quiescence at the end of the first postnatal week in wildtype mice involves a modulation of Sufu function (i.e. another regulator might be normally involved in downregulating Shh signaling at that time). Please modify these interpretations of the data.

7) Why did the authors choose to do their analysis at P7 in Figure 2 and 3? The data from experiments in Figure 4-6 indicate that P7 is a transitional stage in which there is likely a mix of "developmental" and "adult-like" NSCs. Therefore, after going through Figure 4-6, it is unclear if the interpretation of the data at P7 is due to the shift in entry to quiescence or reduced proliferation of adult NSCs. As a result, the data in Figure 2 and 3 seems difficult to interpret and a bit out of place in the manuscript.

8) Overall, the dual thymidine analog experiment is difficult to interpret, because proliferation levels are different between genotypes at P0, 3, and 7, and the proliferation rates at 8 weeks are not presented (see comment #6). For example, if proliferation in adult DG decreased in Sufu KO animals, then IdU in the dual thymidine analog experiment would label fewer cells in Sufu KO animals, which would confound the interpretation of this experiment; the decrease in CldU+ IdU+ cells at P3 and P7 time points could be due to the altered transition of NSCs into quiescence or it could be due to the reduced proliferation in the adult or both. It might be simpler to interpret the results if you focus on CldU+ cells alone and look at how many CldU+ RGLs are generated at each time point in this experiment. On a similar note, most CldU+ IdU+ cells counted in Figure 6D and 6H are DCX+ (Figure 6F and 6I), which means that these data might reflect levels of proliferation and neurogenesis in the adult rather than altered entry into quiescence at postnatal stage. Additionally, the greater CldU+ IdU+ signals at P3 could indicate greater proliferation before P3 or greater quiescence after P3, or both. The gradual entry of NSCs into quiescence could be looked at in a simpler and more convincing fashion by examining the proportion of dividing (Ki67+) Sox2+ NSCs in wildtype mice from P0 to P7.

---

## [Author Response]

Essential revisions:1) The authors show that there is a large expansion of NSCs during normal development of the dentate gyrus between P0 to P7 (Figure 5A-D) and they claim that deletion of Sufu causes NSCs to precociously transition to a quiescent state (P0-3 instead of P3-7 normally), thus reducing the pool of quiescent NSCs in the adult (Figure 4F). However, Figure 2B clearly shows that there is no difference in the number of Sox2+ NSCs in the dentate gyrus of Sufu KO mice at P7. If the expansion of NSCs occurs from P0 to P7, then the decrease in NSC number should be apparent by P7. In agreement with this, the model presented in Figure 7 suggests that there are fewer NSCs (quiescent and dividing combined) in the Sufu KO at P3-7, but the data in Figure 2B clearly shows that there is no difference in the number of Sox2+ NSCs at P7. Along the same line, if there is an excessive proliferation of NSCs in SmoM2 mice (Figure 2G, H), why is their number unchanged at P7 (Figure 2F)? The authors should comment on these apparent contradictions and may need to amend their model.

We understand the reviewers’ concerns. We have considered which stage is appropriate to examine the result of *Sufu* deletion. According to the data in Figure 6 (CldU and IdU), P7 is still a transitional stage in which NSCs shift to a quiescent state from a developmental proliferative state. Therefore, Sox2+ cells in the SGZ of P7 DGs would contain both states of NSCs, and there are still some expanding NSCs in the SGZ of P7 DGs. Given that the transition and expansion of NSCs are not completed yet at P7, it is difficult to determine the overall effects on the NSC pool in the Sufu-KO mice at this stage. Figure 6 demonstrated that the transition to quiescent state concludes by P14. Therefore, we have checked the number of Sox2+ cells at P14 as well (Figure 2 and Figure 3—figure supplement 2A-E). In P14 DGs, we found that the number of Sox2+ cells was significantly reduced in the SGZ of Sufu KO mice. We included this new data in Figures 2, 3 and Figure 3—figure supplement 2.

We also appreciate the helpful comments about Figure 7. We agree that Figure 7 did not comport well the data in the original Figure 2 and 3. So, we have now modified Figure 7, in which Sufu-KO mice have equal amount of total NSCs (quiescent and dividing combined) and less number of dividing NSCs at P7, whereas the number of total NSCs are less at P14 and Adult.

We also understand the reviewers’ concerns about SmoM2 mice. We further analyzed the lack of major increases in the absolute number of Sox2+ NSCs in SmoM2 DG at P7. As shown in Figure 2, there is an increase in proliferating cells as labeled by Ki67 at this stage, in contrast to Sufu-KO mice. Simultaneously, we also found increased cell death as indicated by the expression of cleaved caspase 3+ cells in the DGs of SmoM2 mice at P7 (Figure 2—figure supplement 1). Thus, despite the drastic increase in proliferating NSCs, many of these cells do not survive and this leads to the comparable numbers of Sox2+ NSCs in control and SmoM2 DG. We have now added this information in Figure 2—figure supplement 1 and the revised manuscript.

2) The authors show in Figure 5G, H that Sufu KO causes reduced proliferation of NSCs in the dorsal but not ventral dentate gyrus at P3. However, in Figure 1C it appears as though most of the Shh-responsive cells are in the posterior/ventral dentate gyrus at P7, which is presumably true for P3 as well. Can the authors explain this discrepancy?

We appreciate these comments because they point out the difficulty of visualizing the ventral DG in coronal sections. To help clarify specific DG regions, we have corrected the figures in this revised manuscript to include the areas we designated as posterior/ventral dentate gyrus (see Figure 1—figure supplement 1). Figure 1C of the original manuscript showed coronal sections, which unfortunately did not show images that included the ventral DG. We have now added images that include the entire DG and marked dorsal and ventral DG regions in Figure 1—figure supplement 1. In this figure we showed a higher number of Gli1-LacZ+ cells localized in the dorsal DG compared with the ventral DGs of Sufu^fl/fl^;Gli1^lacZ/+^ mice. This supports our findings that there is a specific reduction in proliferating NSCs in the dorsal but not ventral DG.

3) The authors show in Supplementary Figure 1 that Gli1 deletion has no effect on the number or proliferation of NSCs at P7 in the dentate gyrus and conclude that "Gli1 is largely dispensable [for forebrain development]". If this is true, then can the authors comment on why they think that a Gli1-lacZ reporter line is a good reporter of the level of Shh signaling required for development of the dentate gyrus (Figure 1)? In light of the fact that Sufu KO unexpectedly decreases Gli1-lacZ reporter expression, is there another way to confirm that Sufu KO decreases Shh signaling in the developing dentate gyrus? Related to this, the interpretation of the data on the regulation and function of Gli genes in wildtype and Sufu KO mice in Figure 3G is somewhat confusing and need to be clarified (subsection “In the absence of Sufu, Gli1 function becomes responsible for proper 275 proliferation of NSCs during DG development”).

Thank you for helpful comments about function of Gli1 and Gli1-lacZ mice. We realized that we didn’t adequately discuss Gli1 function in activating Shh signaling and also why the Gli1 promoter has been utilized to reliably detect Shh signaling activity. Gli1 functions to support Gli2 transcription activator function, but is compensated for extensively by Gli2 (Bai et al., 2002; Park et al., 2000). In the absence of Gli1, other Gli activators such as Gli2 are able to compensate and activate Shh signaling target gene expression (Park et al., 2000). Therefore, deletion of Gli1 alone does not cause developmental defects. Instead, it produces severe developmental defects when combined with reduced Gli2 expression (Bai et al., 2002; Park et al., 2000). Nevertheless, Gli1 is a reliable and direct transcriptional target of Gli2. Thus, Gli1 expression is strictly induced upon Shh stimulation (Bai et al., 2002; Lee et al., 1997), making Gli1-LacZ mice a useful readout for Shh signaling activity (Ahn and Joyner, 2005; Balordi and Fishell, 2007; Ihrie et al., 2011; Machold et al., 2003). We also confirmed by in situ hybridization thatexpression of *Ptch1*, a downstream target of Shh signaling, is detected where LacZ+ cells are localized in Gli1-LacZ mice in the developing DG (see Author response image 1 below). We have added further explanation of the Gli1 reporter mice and the relationship of Gli1 and Gli2 in the revised manuscript.

To further support our Gli1-LacZ findings and confirm that Shh signaling is reduced in Sufu-KO mice, we conducted qPCR to determine the expression levels of Shh signaling target genes. We tested the expression of Shh signaling target genes (*Ptch1, N-Myc, CyclinD1 and Cxcr4*) in P3 DGs. We found in Sufu-KO mice, significant reductions were observed in two Shh targets: *CyclinD1* and *Cxcr4* expression. On the other hand, in the Sufu-KO;Gli1^lacZ/+^ mice, significant reductions in the expression of *Ptch1, N-Myc, CyclinD1* and *Cxcr4* were observed (Figure 3—figure supplement 1H). Thus, the reduction in Gli1-LacZ activity in Sufu-KO;Gli1^lacZ/+^ simultaneously and accurately reflected a decrease in Shh signaling activity. We have now added new qPCR data in Figure 3G and Figure 3—figure supplement 2H of the revised manuscript.

In this study, although we could not address the exact mechanism, we believe that reduced expression of both Gli activators (Gli1 and Gli2) may partially explain the greater reductions of Shh signaling target genes expression and profound phenotypes in Sufu-KO;Gli1^lacZ/+^ mice. Indeed, it is demonstrated that reducing Gli2 level in Gli1 homozygous null mice (which develop normally), leads to multiple developmental defects and death soon after birth (Bai et al., 2002; Park et al., 2000). We have included this interpretation in the revised manuscript.

4) Important for the interpretation of the data is the assumption that DG development is not affected until the first postnatal week. The authors show that total Sox2+ cell numbers are unchanged. However, this appears to be a rather superficial read out and should be expanded by other measures (e.g., estimates of total neuronal and glial populations). If Sufu deletion affects the formation of the DG, all effects could be at least be partially explained by secondary effects. Using a conditional Cre line to test later stages of Sufu deletion (e.g., early postnatal) would be one way to address this concern. Related to this, the study focuses on the proliferation and quiescence of NSCs, yet much of the analysis is done by quantifying Sox2+ cells, which include both NSCs and the larger population of IPCs. Shh/Sufu might regulate the proliferation of IPCs as well as NSCs and it is unclear how much the changes in cell proliferation observed in loss and gain of function mutants concern specifically NSCs. The authors should use a more specific NSC marker or reassess their conclusions.

We understand the concern that Sufu deletion by hGFAP-Cre may impair DG development before postnatal stages. In the revised manuscript, we show that there is no difference in the proliferation of NSCs at P0 (Figure 5E and Figure 6—figure supplement 2A and B). Therefore, we believe that prenatal DG development is not impaired by deletion of Sufu. We also have tested the effect of *Sufu* deletion temporally using Gli1^CreER/+^ mice by specifically inducing Sufu deletion at E17.5 (Gli1^CreER/+^;Sufu-KO; Ai14). In this mouse, Cre recombination (indicated by expression of Ai14) is induced in NSCs in the ventral hippocampus, which later migrate to the DG and become adult NSCs as we showed in Figure 5. Thus, we believe that Sufu deletion is induced specifically in NSCs in this mouse model. This would minimize the secondary effects and potential developmental abnormalities at prenatal stages. We analyzed these mice at P7 and found that the number of Sox2+/Ai14+ cells and Sox2+/Ki67+/Ai14+cells were significantly reduced in Gli1^CreER/+^;Sufu^fl/fl^;Ai14 mice compared with Gli1^CreER/+^;Sufu^fl/+^;Ai14. These findings are consistent with the phenotypes observed in the hGFAP-Cre mice. We have now added these data to Figure 5—figure supplement 1A-C.

As suggested by the reviewers, we investigated the effects of *Sufu* deletion on IPC production and proliferation in Sufu-KO mice. We found that the number of Tbr2+ IPCs was significantly reduced in Sufu-KO mice (Figure 3E, F and Figure 3—figure supplement 2D-G). Further assessments showed that the reduction in Tbr2+ cells in Sufu-KO mice is not due to impaired or reduced proliferation of IPCs since we found no differences in the Ki67+ proliferating cell population in Tbr2+ IPCs between control and Sufu-KO mice Figure 3—figure supplement 2F. These data suggest that deletion of *Sufu* does not affect the proliferation of IPC. Rather, our analyses show that deficits in IPC numbers were due to the failure of NSCs to generate this cell population. Thus, we believe that reduction of the proliferating cell population in Sox2+ in Sufu-KO mice reflects reduction of NSC proliferation. We have added these results in Figure 3 and Figure 3—figure supplement 2.

5) In Figure 4, the authors analysed neurogenesis in the adult DG and found reduced numbers of radial glia-like NSCs and reduced rate of neuron production, however it is unclear whether NSC proliferation is altered (percentage of NSCs that are Ki67+) and whether the total number of DCX+ cells produced decreases in adult Sufu KO mice.

Thank you for helpful suggestions. We have now added data examining the percentage of Ki67+ cells in Sox2+ NSCs in Figure 4—figure supplement 1C and D, in which there is no difference in the proliferating cell population of NSCs in adult DG between control, Sufu-KO and Sufu-KO;Gli1^lacZ/+^ mice. We have also investigated the number of total DCX+ cells and Tbr2+ cells (Figure 4D and Figure 4—figure supplement 1A and B). We found that DCX+ cells and Tbr2+ cells were significantly reduced in Sufu-KO;Gli1^lacZ/+^ mice. We have added these data in Figure 4 and Figure 4—figure supplement 1.

6) The authors overinterpret their data regarding the role of Sufu in regulation of Shh signaling and NSC quiescence. The last sentence of the Introduction states that "Sufu (…) is important for ensuring (…) the timely transition to a quiescent state during DG development". However there is no evidence that Sufu is implicated in determining the timing of the transition to quiescence. The authors add at the beginning of the Discussion: "Thus Sufu modulates the timing of quiescence of NSCs in DG development by controlling NSCs expansion via modulation of Shh signaling activity". Again the paper does not provide evidence that Sufu normally modulates Shh activity and regulates the timing of quiescence of NSCs. Changes in Shh signaling and in timing of NSC quiescence are observed in Sufu mutant mice but there is no evidence that entry of NSCs in quiescence at the end of the first postnatal week in wildtype mice involves a modulation of Sufu function (i.e. another regulator might be normally involved in downregulating Shh signaling at that time). Please modify these interpretations of the data.

We have modified our statements to clearly reflect our interpretations of how Sufu plays a role in regulating NSC quiescence. As the reviewers point out, there is no direct evidence that Sufu regulates the timing. Additionally, we have included in the Discussion other factors previously reported to play a role in regulating the production and timing of qNSCs in the DG. Whether or not Sufu functions in concert with these factors is not yet known and could be an interesting future direction.

7) Why did the authors choose to do their analysis at P7 in Figure 2 and 3? The data from experiments in Figure 4-6 indicate that P7 is a transitional stage in which there is likely a mix of "developmental" and "adult-like" NSCs. Therefore, after going through Figure 4-6, it is unclear if the interpretation of the data at P7 is due to the shift in entry to quiescence or reduced proliferation of adult NSCs. As a result, the data in Figure 2 and 3 seems difficult to interpret and a bit out of place in the manuscript.

This is a reasonable point. P7 is not necessarily the optimal timing to test the effects of *Sufu* deletion since it is such a transitional time in dentate formation; to get a more full picture it would be helpful to examine a slightly more advanced stage. Therefore, we have also analyzed the DG at P14, in which transition of "developmental" and "adult-like" NSCs seems largely complete. At this time point, we found that Sox2+ cells were significantly reduced in the DGs of Sufu-KO mice. Thus, P14 is an appropriate time to detect any defects arising from impaired expansion of the NSC pool. We have added these P14 data in the revised manuscript (Figure 2 and Figure 3—figure supplement 2).

8) Overall, the dual thymidine analog experiment is difficult to interpret, because proliferation levels are different between genotypes at P0, 3, and 7, and the proliferation rates at 8 weeks are not presented (see comment #6). For example, if proliferation in adult DG decreased in Sufu KO animals, then IdU in the dual thymidine analog experiment would label fewer cells in Sufu KO animals, which would confound the interpretation of this experiment; the decrease in CldU+ IdU+ cells at P3 and P7 time points could be due to the altered transition of NSCs into quiescence or it could be due to the reduced proliferation in the adult or both. It might be simpler to interpret the results if you focus on CldU+ cells alone and look at how many CldU+ RGLs are generated at each time point in this experiment. On a similar note, most CldU+ IdU+ cells counted in Figure 6D and 6H are DCX+ (Figure 6F and 6I), which means that these data might reflect levels of proliferation and neurogenesis in the adult rather than altered entry into quiescence at postnatal stage. Additionally, the greater CldU+ IdU+ signals at P3 could indicate greater proliferation before P3 or greater quiescence after P3, or both. The gradual entry of NSCs into quiescence could be looked at in a simpler and more convincing fashion by examining the proportion of dividing (Ki67+) Sox2+ NSCs in wildtype mice from P0 to P7.

We agree that decreased neurogenesis might make the analysis using CldU and IdU more confusing. We have also analyzed how many CldU+ RGLs are generated at each time point in this experiment (Figure 6—figure supplement 1). We counted the number of CldU+/Sox2+/GFAP+ radial NSCs and found a similar trend to what we observed in CldU/IdU experiments; the number of CldU+/Sox2+/GFAP+ radial NSCs was decreased significantly in P3 and P7 CldU-injected groups, whereas it was increased significantly in P0 CldU injected groups in Sufu-KO and Sufu-KO;Gli1^lacZ/+^ mice.

Following the reviewers’ suggestion, we also examined the proportion of dividing (Ki67+) Sox2+ NSCs from P0 to P7. We found that the proliferating cell population was comparable in P0 DG. However, it was decreased at P3 and P7 in the SGZ of Sufu-KO and Sufu-KO;Gli1^lacZ/+^ mice. These data indicate that deletion of *Sufu* prematurely decreases the proliferation of NSCs. We have added those results in Figure 6—figure supplement 2.